# Transcription factor *TFCP2L1* patterns cells in the mouse kidney collecting ducts

Max Werth[1†], Kai M Schmidt-Ott[1,2,3†], Thomas Leete[1], Andong Qiu[1,4], Christian Hinze[2], Melanie Viltard[1,5], Neal Paragas[1,6], Carrie J Shawber[1], Wenqiang Yu[1,7], Peter Lee[1], Xia Chen[1], Abby Sarkar[1], Weiyi Mu[1], Alexander Rittenberg[1], Chyuan-Sheng Lin[1], Jan Kitajewski[1,8], Qais Al-Awqati[1], Jonathan Barasch[1*]

[1]Columbia University, New York, United States; [2]Max Delbruck Center for Molecular Medicine, Berlin, Germany; [3]Department of Nephrology and Intensive Care Medicine, Charité - Universitaetsmedizin Berlin, Berlin, Germany; [4]Tongji University, Shanghai, China; [5]Institute for European Expertise in Physiology, Paris, France; [6]University of Washington, Seattle, United States; [7]Fudan University, Shanghai, China; [8]University of Illinois at Chicago, Chicago, United States

**Abstract** Although most nephron segments contain one type of epithelial cell, the collecting ducts consists of at least two: intercalated (IC) and principal (PC) cells, which regulate acid-base and salt-water homeostasis, respectively. In adult kidneys, these cells are organized in rosettes suggesting functional interactions. Genetic studies in mouse revealed that transcription factor *Tfcp2l1* coordinates IC and PC development. *Tfcp2l1* induces the expression of IC specific genes, including specific $H^+$-ATPase subunits and *Jag1*. *Jag1* in turn, initiates Notch signaling in PCs but inhibits Notch signaling in ICs. *Tfcp2l1* inactivation deletes ICs, whereas *Jag1* inactivation results in the forfeiture of discrete IC and PC identities. Thus, *Tfcp2l1* is a critical regulator of IC-PC patterning, acting cell-autonomously in ICs, and non-cell-autonomously in PCs. As a result, *Tfcp2l1* regulates the diversification of cell types which is the central characteristic of 'salt and pepper' epithelia and distinguishes the collecting duct from all other nephron segments.

*For correspondence: jmb4@columbia.edu

[†]These authors contributed equally to this work

Competing interests: The authors declare that no competing interests exist.

## Introduction

Kidney development starts from two embryonic structures, the metanephric mesenchyme (MM) and the ureteric bud (UB) (*Little and McMahon, 2012*). The Metanephric Mesenchyme gives rise to tubules organized in segments, each with a single type of epithelial cell. The UB, in contrast, gives rise to tubules containing mixtures of two different paradigmatic cell types called principal cells (PC) and intercalated cells (IC). These two cell types have different functions and they express different sets of signature proteins. PCs control water and electrolyte balance and express ion-channels such as ROMK, ENaC and Aquaporins 2–4 (*Sasaki et al., 1995*). ICs, on the other hand, control acid-base balance (*Gluck et al., 1982*) and contribute to immune defense (*Paragas et al., 2014*; *Azroyan et al., 2015*). ICs express vacuolar $H^+$-ATPase, including kidney specific subunits (D-B subunits), bicarbonate conversion enzymes (carbonic anhydrase *Ca2* and *Ca12*), bicarbonate transporters, AE1 (*Slc4a1*), AE4 (*Slc4a4*) and pendrin (*Slc26a4*). Nonetheless, despite their many differences, ICs and PCs function coordinately: for example, the absorption of $Na^+$ by PCs creates a transepithelial electrical gradient which stimulates $H^+$ secretion by ICs (*Chang et al., 1996*; *DuBose and Caflisch, 1985*).

The underlying mechanisms that allow ICs and PCs to coordinate their activities have been difficult to identify because their genesis from progenitors and their ultimate relatedness to one another

has been uncertain. This is because the collecting duct is thought to be populated by many subtypes of ICs, including α-IC and β-IC, non-α-β-IC, and various mixed cell types (*Schwartz et al., 1985*) interspersed among PCs. Yet the definition of these various ICs and their relationship to PCs has been called into question as one cell type may convert into another in response to environmental challenges and recently reproduced in genetic models (*Aigner et al., 1995*; *Al-Awqati and Schwartz, 2004*; *Bagnis et al., 2001*; *Wu et al., 2013*; *Trepiccione et al., 2016*). Moreover, the knockout of the IC specific transcription factor, *Foxi1* exhibited a cell type that co-expressed mixtures of IC (*Ca2*) and PC (*Aqp2*) proteins, implying that *Foxi1* restricted the otherwise facile interconversions of ICs to PCs (*Blomqvist et al., 2004*). Consequently, while many distinct cellular phenotypes are known to populate the collecting duct, the underlying logic that coordinates these cell types has not been uncovered.

A clue to the mechanisms that coordinate ICs and PCs was suggested by their stereotyped spatial patterning. Immunofluorescence analysis found rosette-like structures in the adult collecting duct, a pattern reminiscent of tissues governed by Notch mediated lateral inhibition (*Blanpain et al., 2006*; *Kiernan, 2013*; *Noah et al., 2013*). In fact, recent studies have shown that manipulation of Notch signaling modifies the ratio of PCs and ICs (*Jeong et al., 2009*; *Guo et al., 2015*; *Grimm et al., 2015*; *Nam et al., 2015*) suggesting that not only *Foxi1* but also components of the Notch pathway are critical to determine cell type. However, the developmental context for these regulators is currently indeterminate, in part because of incomplete description of the developmental origin of IC and PC.

Here we show that IC-PC coordination is under control of a poorly studied transcription factor called *Tfcp2l1*. We found that *Tfcp2l1* induces the initial formation of a cellular intermediate which we call the 'double positive' mixed IC-PC cell. Thereafter *Tfcp2l1* regulates the formation of discrete ICs and PCs by both cell-autonomous and cell non-autonomous mechanisms. The latter includes the regulation of the *Jag1-Notch1* pathway in rosettes composed of ICs and PCs. These data indicate that UB tubules are patterned by Notch dependent interactions of neighboring cells rather than demarcated in nephron segments controlled by Notch signaling (*Costantini and Kopan, 2010*).

In sum, coordinate development of ICs and PCs is linked by *Tfcp2l1* acting late in gestation in progenitors of the collecting duct. This mechanism explains the apparent reciprocal relationship in the relative abundance of ICs and PCs in the adult collecting duct (*Jeong et al., 2009*; *Guo et al., 2015*) as well as their physiologic linkage. We suggest that coordination between ICs and PCs by *Tfcp2l1* is critical for homeostasis, since these cells co-regulate the balance of electrolytes, acid-base, and water.

## Results

### Expression of *Tfcp2l1* in the development of the distal nephron

*Tfcp2l1* (also known as LBP-9 or CRTR-1) is a nuclear transcription factor and a member of the CP2 subfamily of the LSF/Grainhead family (*Kokoszynska et al., 2008*; *Yamaguchi et al., 2005*; *Aue et al., 2015*; *Werth et al., 2010*; *Walentin et al., 2015*; *Traylor-Knowles et al., 2010*). *Tfcp2l1* has been implicated in the maintenance of pluripotency networks of ES cells where it is targeted by both LIF (*Dunn et al., 2014*; *Martello et al., 2013*; *Ye et al., 2013*) and Wnt (*Yamaguchi et al., 2005*). In addition, *Tfcp2l1* is implicated in the development of arborizing epithelial trees, including the collecting ducts (*Yamaguchi et al., 2006*; *Paragas et al., 2014*). In fact, *Tfcp2l1* was detected at E11 in the primordium of the collecting ducts (the Wolffian Duct and the Ureteric Bud; data not shown), and then throughout its arborized $Krt8^+$ stalks at E15-E18 (*Figure 1A*, *Figure 1—figure supplement 1*) when *Tfcp2l1* appeared to localize exclusively to the nucleus. In adult collecting ducts (P60), *Tfcp2l1* was prominent in both PCs ($Krt8^+$) and ICs ($Atp6v1b1^+$; abbreviated *Atp6b1*) (*Figure 1B*), but demonstrated greater immunoreactivity in ICs than in PCs (*Figure 1C*). *Tfcp2l1* was also expressed in the Thick Limb of Henle and connecting segments of the nephron (data not shown), but its most persistent location was the collecting duct system.

### Identification of IC-PC 'double positive' progenitors

In order to identify the cellular targets of *Tfcp2l1*, the development of collecting duct epithelia was characterized using signature proteins found in adult PCs and ICs. While the E13 and E15 collecting

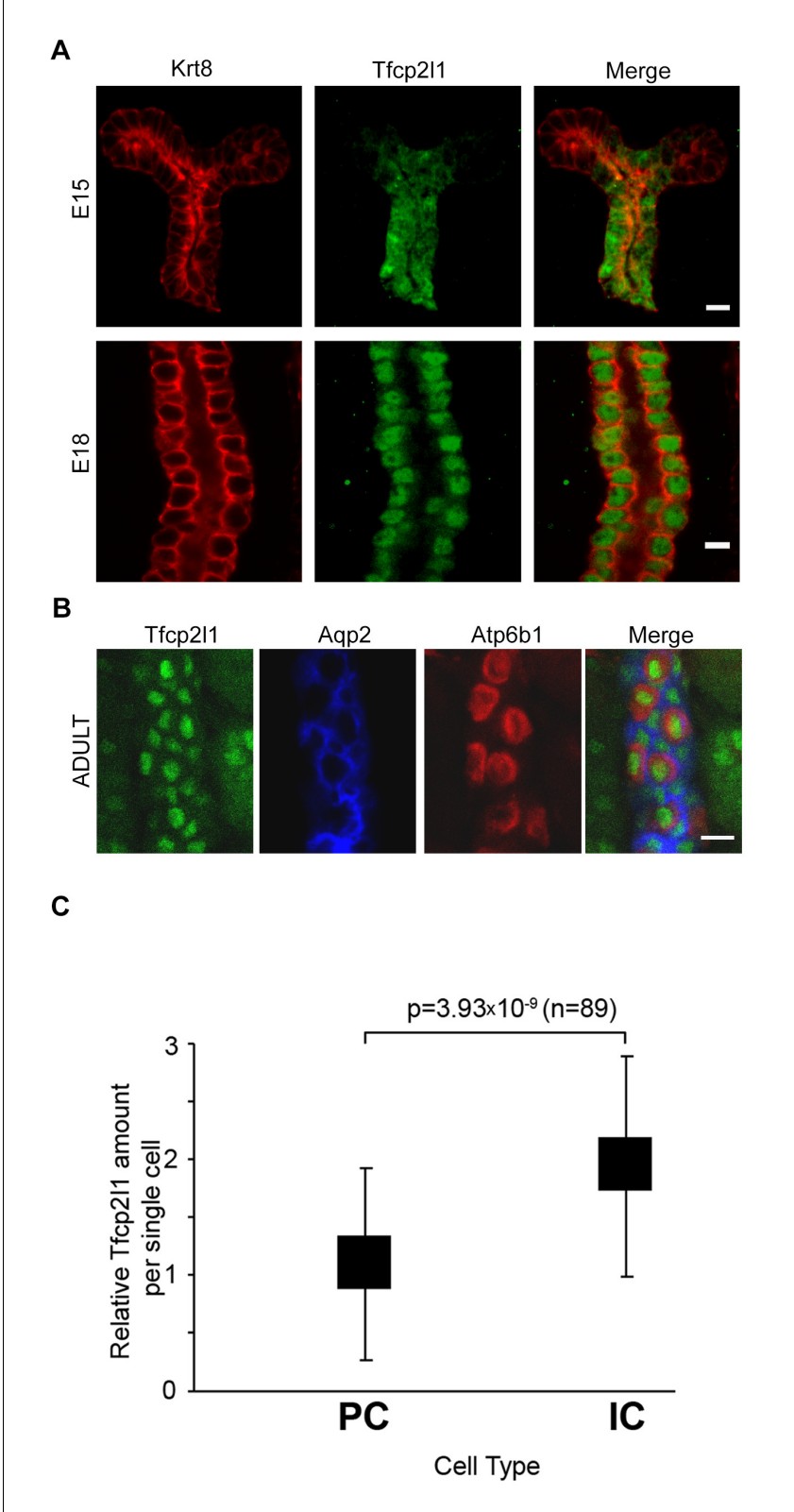

**Figure 1.** *Tfcp2l1* is a nuclear protein expressed in the collecting ducts. (**A**) Immunofluorescence detection of *Tfcp2l1* (green) in stalks of ureteric-collecting ducts at E15 and at E18. Nuclear localization was prominent at E18. The ducts were identified by the uniform expression of *Krt8* (red). Bars = 5 μm. (**B**) In adult collecting ducts, *Tfcp2l1* (green) was expressed by both Intercalated Cells (IC), identified by immunodetection of *Atp6v1b1*, abbreviated

*Figure 1 continued*

*Atp6b1* (red), and Principal Cells (PC) identified by immunodetection of *Aqp2* (blue). Z-stack projection. Bar = 10 μm. (C) Quantification of *Tfcp2l1* immunoflourescence in adult collecting ducts normalized per measurement area. *Atp6b1*[+] IC cells expressed higher levels of *Tfcp2l1* than did *Aqp2*[+] PC cells.

The following figure supplement is available for figure 1:

**Figure supplement 1.** Expression of *Tfcp2l1* message in the stalks of the ureteric bud (E15 mouse kidney; in-situ hybridization).

duct demonstrated homogeneous expression of proteins typical of PCs, there were rare cells co-expressing IC proteins (not shown). By E18 these cells became abundant and we designated them 'double-positive' progenitors, because they co-expressed IC and PC proteins (e.g. IC proteins: *Foxi1*[+], *Atp6b1*[+]; together with PC proteins: *Calb1*[+], *Krt8*[+]: *Figure 2A*, *Figure 2—figure supplement 1A,B*). The 'double positive' phenotype was spatially restricted, because neighboring cells expressed PC but not IC proteins (*Figure 2—figure supplement 1A,B*). The 'double-positive' progenitors were also transient; by birth, IC and PC proteins were generally not co-expressed (*Figure 2—figure supplement 1C*, *Figure 3*, *Figure 3—figure supplement 2*; four independent mice for each immunofluorescence analysis) except in rare collecting duct cells (*Figure 2—figure supplement 2*).

To further analyze the origin of ICs and PCs, we crossed cre-reporter (*mTmG*; *Muzumdar et al., 2007*) with cre-drivers that become active at different developmental stages of collecting duct development. We used *HoxB7-Cre*, which becomes active before the appearance of 'double positive' cells, and *Atp6b1-Cre*, which is active only after the expression of *Atp6b1* in ICs between E15-E18 (*Miller et al., 2009*). All ICs and PCs were labeled in *HoxB7-Cre/mTmG* adult kidneys (P60; *Figure 2B*) confirming that ICs and PCs developed from a *HoxB7*[+] ureteric progenitor cell. Surprisingly, *Atp6b1-Cre/mTmG* labeled not only *Atp6b1*[+] ICs (74% of *GFP*-labeled cells were ICs) but also a subset of *Aqp2*[+] PCs (24% of *GFP*-labeled cells were *Aqp2*[+] cells expressing low levels of *Atp6b1*). In addition, rare (2%) *GFP*-labeled cells expressed equivalent levels of *Aqp2* and *Atp6b1*, typical of 'double positive' cells (n = 80 ducts in four independent mice were inspected for each genetic label). Therefore, all adult ICs and PCs derive from *HoxB7*[+] progenitors and all adult ICs and at least some PCs derive from *HoxB7*[+] progenitors that subsequently express *Atp6b1*[+].

## *Tfcp2l1* is required for epithelial patterning

To determine whether *Tfcp2l1* is important for the development of 'double-positive' progenitors or for later stages of IC and PC development, we created a floxed allele flanking the CP2 domain (*Figure 3a*), and we used a set of cre-drivers to inactivate *Tfcp2l1* in a stage- and cell-specific manner. *EIIA-Cre* was used to delete *Tfcp2l1* early in development (*Lakso et al., 1996*), the *Cdh16-Cre* driver was used to inactivate *Tfcp2l1* throughout the distal nephron and collecting duct before ICs develop (*Shao et al., 2002*), and *Atp6b1-Cre* was used for cell-specific deletion in maturing ICs (*Miller et al., 2009*). In each case, the efficiency of *Tfcp2l1* inactivation was confirmed by immunostaining (*Figure 3B*). *EIIA-Cre;Tfcp2l1^{f/f}* knockouts died quickly after birth when IC maturation was still ongoing (similar to published gene-trap; *Yamaguchi et al., 2006*), but both *Cdh16-Cre* and *Atp6b1-Cre;Tfcp2l1^{f/f}* knockouts survived to adulthood (P60).

*Tfcp2l1* deletion with *EIIA-Cre* or with *Cdh16-Cre* produced grossly normal kidneys at birth (*Figure 3—figure supplement 1*), except that all 'double positive' progenitors and all mature ICs were deleted, as measured by their signature proteins, *Foxi1*, *Ca2*, and *Atp6b1* which for example were reduced by ~89% (4.8 ± 1.9 *vs* 41.5 ± 15.1 *Atp6b1*[+] cells per kidney section; n = 3 independent mice; p=0.0014; *Figure 3B*; *Figure 3—figure supplement 2*). In contrast, PC cell numbers and proteins demonstrated variable responses and were either mildly downregulated (e.g. 78 ± 23.6 *vs* 112.5 ± 22.6 *Aqp2*[+] cells per kidney section; n = 3 independent mice; p=0.04; *Figure 3B*) or mildly upregulated (e.g. *Krt8*; *Figure 3—figure supplement 2A*). PCs also remained correctly polarized, displaying apical ROMK and basolateral *Cdh1* and *Aqp4*, indicating that *Tfcp2l1* did not regulate epithelial polarity in PCs (not shown). Hence, *Tfcp2l1* was critical to initiate the expression of IC signature genes which were first expressed in 'double positive' progenitors.

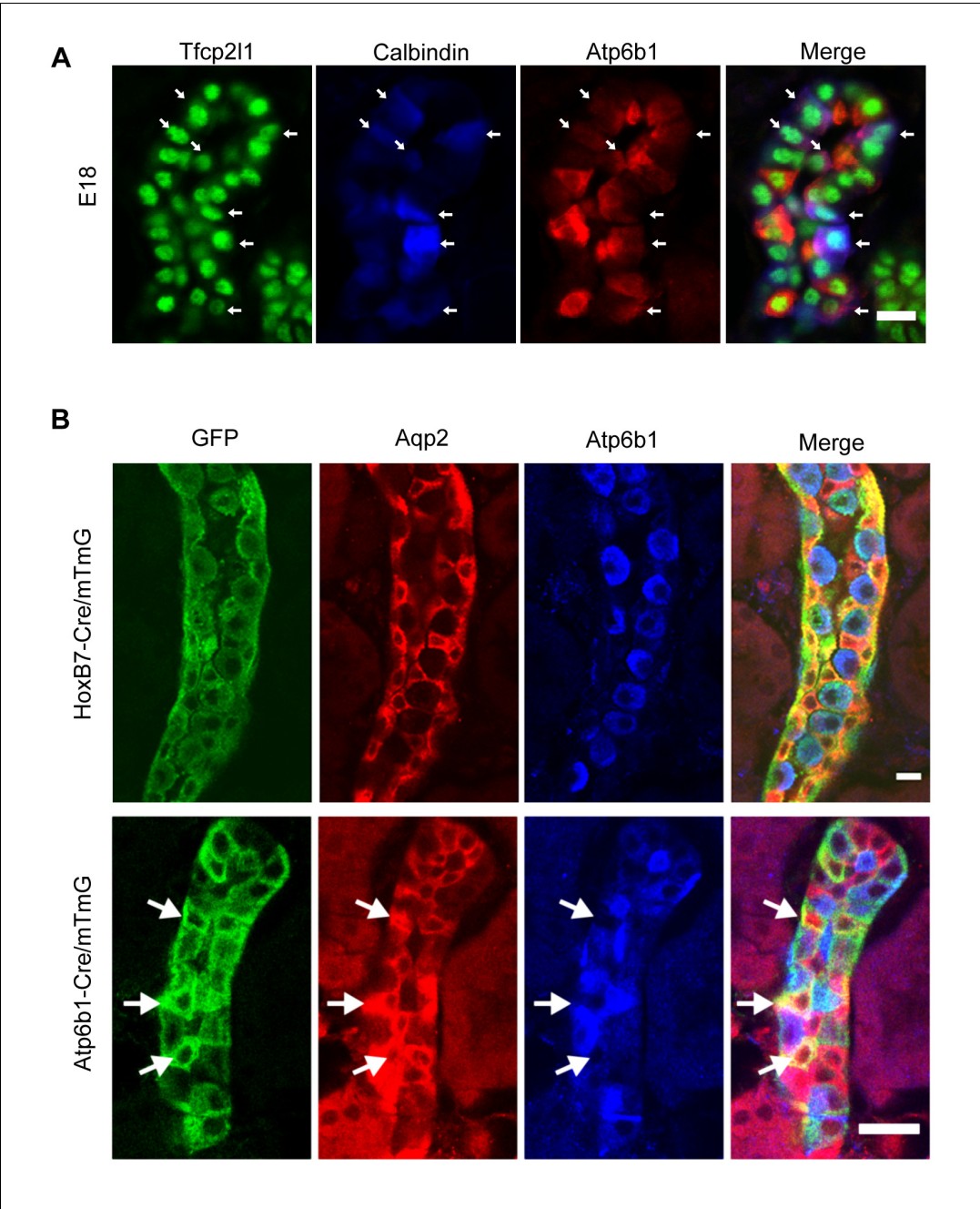

**Figure 2.** 'Double positive' progenitors populate the E18 collecting duct. (**A**) IC proteins were expressed at E18 in presumptive PC cells. Co-expression of *Atp6b1* (typical of ICs, red) and Calbindin (typical of PCs, blue) is shown in *Tfcp2l1*$^+$ cells in the cortical region of the collecting duct. (white arrows) Bar = 10 µm. (**B**) Lineage of ICs and PCs was detected with genetic reporters. *HoxB7-Cre;mTmG* (green) marked every cell in the collecting duct including *AQP2*$^+$ PC cells (red-yellow) and *Atp6b1*$^+$ IC cells (blue-green). *Atp6b1-Cre;mTmG* labeled every IC cell (endogenous *Atp6b1*$^+$; blue-green), as well as some *Aqp2*$^+$ PCs (white arrows, yellow). *Atp6b1-Cre;mTmG*-negative PC cells are also found (*Aqp2*$^+$, *Atp6b1*$^-$, *Atp6b1-Cre;mTmG*$^-$, red) (Bars= top 5 µm, bottom 20 µm).

The following figure supplements are available for figure 2:

**Figure supplement 1.** Detection of 'double positive' precursors in the embryonic collecting ducts.

**Figure supplement 2.** Detection of rare 'double positive' cells in adult collecting duct using marker proteins Atpb1 (IC cells) and *Aqp2* (PC cells).

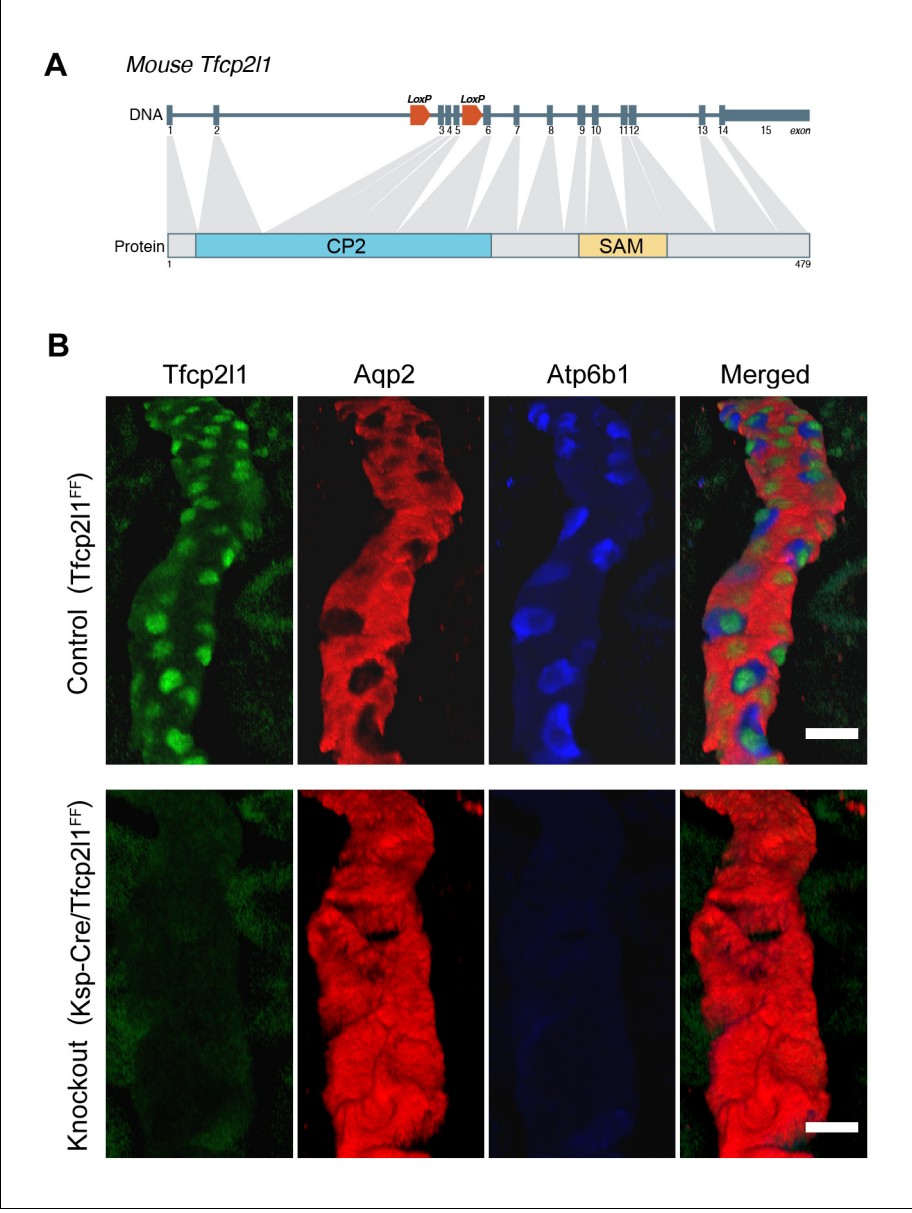

**Figure 3.** *Tfcp2l1* is necessary for the development of ICs. (**A**) Structure of mouse *Tfcp2l1* gene showing LoxP sites flanking the DNA-Binding CP2 domain (exon 3 and 4). (**B**) Control (*Tfcp2l1*$^{f/f}$) and *Tfcp2l1* knockout kidneys (*Cdh16-Cre;Tfcp2l1*$^{f/f}$) were analyzed for PC (*Aqp2* red) and IC (*Atp6b1* blue) proteins. Note that the deletion of *Tfcp2l1* replaced the normal patterning of IC and PC cells with a monotonous array of PC like cells (*Aqp2*$^{+}$). Z-stack reconstruction. Bars = 10 μm.

The following figure supplements are available for figure 3:

**Figure supplement 1.** Gross kidney morphology was preserved after the global deletion of *Tfcp2l1* (*EIIA-Cre; Tfcp2l1*$^{f/f}$).

**Figure supplement 2.** *Tfcp2l1* is necessary for the development of ICs.

In contrast to the deletion of *Tfcp2l1* by *EIIA-Cre* and *Cdh16-Cre* drivers which act before the appearance of IC cells, we found that deletion of *Tfcp2l1* with *Atp6b1*Cre resulted in a normal number and distribution of ICs and PCs at P14. By P60, however, there were fewer than half the number of ICs (14 ± 5.4 *vs.* 29 ± 13 *Atp6b1*+ cells per section or as a percentage of collecting duct cells; n = 3 independent kidneys each; p=0.01; *Figure 4A*). In fact, the surviving ICs demonstrated only faint *Atp6b1* staining which was co-expressed with PC protein *Krt8*. When we introduced the floxed *mTmG* Reporter with *Tfcp2l1*-flox and *Atp6b1-Cre*, and focused our analysis on GFP+ cells by measuring endogenous markers in single cells by spot imaging, we found that GFP+*Tfcp2l1*+ cells demonstrated IC (*Atp6b1*>>*Aqp2*) or PC (*Aqp2*>*Atp6b1*) phenotypes, but GFP+ *Tfcp2l1*-deleted cells appeared strikingly similar to PCs (*Aqp2*>*Atp6b1*; *Figure 4B*) or double positive cells (*Atp6b1* ≈ *Aqp2*).

Taken together, *Tfcp2l1* was critical in the primary development of 'double positive' progenitors and the long-term maintenance of mature ICs which are otherwise capable of assuming a mixed or even a PC cell phenotype. In contrast, *Tfcp2l1* was not likely to mediate IC development by regulating their survival, because we failed to find the apoptosis marker activated caspase 3 in any of our models (data not shown).

## Identification of *Tfcp2l1* target genes

To identify transcriptional targets of *Tfcp2l1* that might explain IC regulation, we used an integrative approach that combined high-throughput data from two independent assays: (1) We found 843 differentially expressed genes in a comparison of *EIIA-Cre;Tfcp2l1*f/f knockout and *Tfcp2l1*f/f control P1 mouse kidneys (Affymetrix Microarrays analyzed by limma, n = 3, p<0.05). According to GUDMAP, the comprehensive genitourinary development database (*Harding et al., 2011*, www.gudmap.org), our analysis showed that 62% of downregulated genes were expressed either by UB, cortical or medullary collecting ducts or by the TALH including multiple subunits of the H+ATPase complex (*Atp6v0d2*; *Atp6b1*; *Atp6v1c2*) and IC specific *Foxi1*, *Ca12*, *Ca2*, *Aqp6* and *Oxgr1* (*Figure 5A*, *Figure 5—figure supplements 1* and *2*; *Supplementary file 1*); (2) We found 6564 *Tfcp2l1*-bound genomic regions in P1 mouse kidney using ChIP-Seq (*Figure 5A*; *Supplementary file 2*). Comparing the two datasets we found that *Tfcp2l1* bound both up and down regulated genes at the TSS (*Figure 5A,B*). In fact, in a global analysis of all *Tfcp2l1* peaks, 30% accumulated near the TSS within known promoters (*Figure 5C*). In addition these peaks demonstrated the established *Tfcp2l1* motif (*Supplementary file 4*) (*Chen et al., 2008*). Moreover, cell type specific actions of *Tfcp2l1* were suggested in a comparison with *Tfcp2l1* ChIP-Seq from ES (*Chen et al., 2008*) and from *Grhl2* ChIP-Seq in kidney (*Werth et al., 2010*) which revealed little overlap with *Tfcp2l1* ChIP-Seq in kidney (*Figure 5D*). In contrast, a random set of genes that were not modulated by the *Tfcp2l* failed to show TSS enrichment (*Figure 5B*).

The comparison between the two datasets also allowed us to identify the critical transcriptional targets of *Tfcp2l1*. Since *Tfcp2l1* is known as an activator (*To et al., 2010*), we sought genes that were significantly downregulated in knockouts and contained a *Tfcp2l1*-DNA interaction site ±50 KB from the gene's transcriptional start site. Using this approach, we found that *Tfcp2l1* positively regulated expression of many IC genes including subunits B1 and D2 of the V-ATPase complex, *Oxgr1*, *Ca12*, *Slc4a1*, *Aqp6* and IC-specific transcription factor *Foxi1* (*Figure 5*, *Figure 5—figure supplement 2*). In contrast, no known PC markers were represented in the downregulated gene set.

It has also been reported that *Tfcp2l1* is a repressor (*Rodda et al., 2001*). Consequently we analyzed genes that were upregulated in the knockouts and that demonstrated a *Tfcp2l1*-DNA interaction site ±50 KB from the gene's TSS. None of these genes were associated with a particular function or a cell-type in the distal nephron.

Taken together, *Tfcp2l1* directly controls genes that are critical for the function of IC cells (differentiation genes, V-ATPase, *Oxgr1* etc).

## *Tfcp2l1* regulates *Jag1* and Notch signaling in the collecting duct

The analysis of our two datasets demonstrated significant over-representation of components of the Notch pathway suggesting that *Tfcp2l1* might regulate Notch signaling in ICs or in PCs.

We analyzed the Notch pathway using well characterized antibodies for Notch Ligands (*Jag1*, *Jag2*, *Dll1*, *Dll4*), Receptors (*Notch1-4*) and Notch Signaling States ('ON': characterized by detection

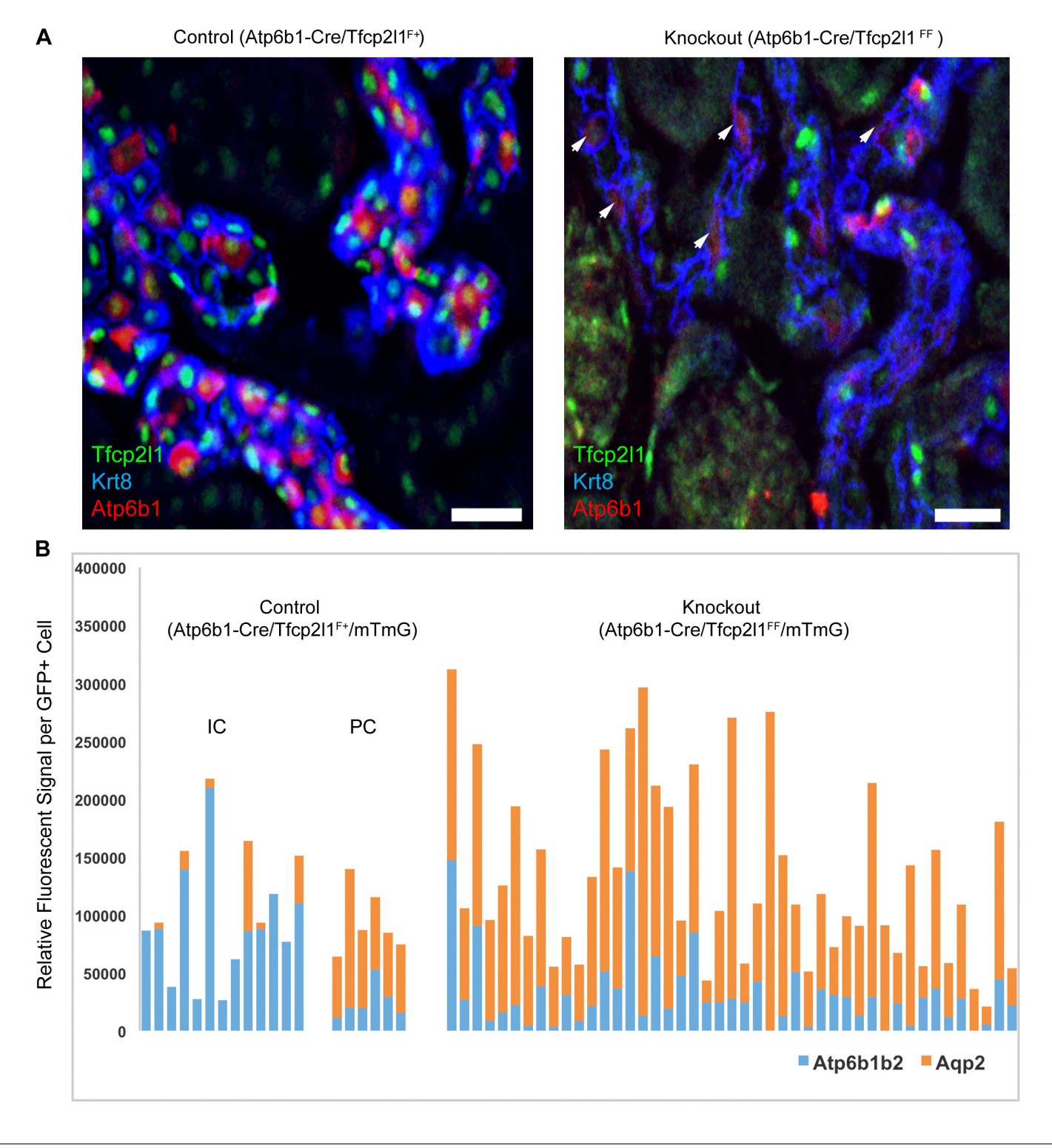

**Figure 4.** IC specific deletion of *Tfcp2l1* (green) by *Atp6b1*Cre results in loss of IC cells. (A) Deletion of *Tfcp2l1* resulted in the widespread loss of IC and PC patterning. Only residual expression of *Atp6b1* (red) in *Krt8*[+] (blue) PCs was detected in cortical collecting ducts (white arrows; n = 3 independent mice; Bars = 25 μm). (B) Cell fate analysis of *Tfcp2l1* knockout IC cells using genetic reporter (*Atp6b1-Cre;mTmG*). We analyzed single *GFP*[+] cells in Control (*Atp6b1-Cre;Tfcp2l1^{f/+};mTmG*) and in Knockout (*Atp6b1-Cre;Tfcp2l1^{f/f};mTmG*) collecting ducts by spot imaging. In control kidneys, *GFP*[+] cells were ICs or PCs (e.g. *GFP*[+] ICs: *Atp6b1>Aqp2* and *GFP*[+] PCs: *Aqp2>Atp6b1*), or expressed both markers in variable ratios

*Figure 4 continued on next page*

*Figure 4 continued*

(*Atp6b1≈Aqp2*). In contrast, in knockout kidneys, the majority of *GFP*+ cells appeared to be PC-like or double positives cell types (compare knockout with wild type profiles). (n = 20 *GFP*+ Control and n = 45 *Tfcp2l1* deleted *GFP*+ cells from representative images; n = 4 independent kidneys).

of cleaved Notch and its target, nuclear *Hes1,* and 'OFF': characterized by detection of uncleaved Notch). We found that *Jag1* marked the development of ICs: *Jag1* and *Atp6b1* co-localize with Ktr8 in IC-PC 'double positive cells' at E18 (*Figure 6A*) and in the adult *Jag1* was specifically expressed by a subset of ICs called Pendrin+ *β*-ICs (*Figure 6B*). The localization of *Jag1* was specific because, α-ICs and PCs were *Jag1*- (*Figure 6C*). In contrast, activated *Notch1* (NICD; Val1744) and activated nuclear *Hes1* were found in PCs (*Jag1*-,*Atp6b1*-,*Aqp2*+) immediately surrounding the *Jag1*+ *β*-ICs (*Figure 7*). Inactive, uncleaved *Notch1* (extracellular domain) was detected in other locations such as the basal membrane of some ICs, identified as *Krt8*-, *Jag1*-α-ICs (*Figure 8* and additional data, not shown). Therefore, the expression of *Jag1* and Notch activity correlated with the development of cell types: PCs ('Notch ON': *Jag1*-, Cleaved Notch1), *β*-IC ('Notch OFF' *Jag1*+, *Notch1*-), and α-IC ('Notch OFF' *Jag1*-, Uncleaved Notch1).

To determine whether *Tfcp2l1* regulated the distinctive patterning of *Jag1* and *Notch1* expression, we examined *Cdh16-Cre;Tfcp2l1*f/f kidneys and found nearly complete deletion of *Jag1* in ICs and nuclear *Hes1* in PCs. In addition, Notch staining was deleted in the *Tfcp2l1* knockout collecting duct (*Figure 8*; n = 4 independent mice). These experiments suggested that *Jag1* and *Notch1* signaling were dependent on *Tfcp2l1*.

To determine whether *Tfcp2l1* might directly control Notch signaling, we examined ChIP analysis and found that *Tfcp2l1* directly bound the promoter of 18 Notch associated genes including *Jag1* (-904, -510), *Hes1* (-56202, -1823, 50028), *Hey1* (-127494, -336, 138049) and *Hey2* (-212) (*Supplementary file 3*). To broaden this analysis, we examined *Tfcp2l1*-bound genomic regions for the enrichment of transcription factor motifs adjacent to the *Tfcp2l1* binding site (PWMErich using top 1000 *Tfcp2l1*-bound regions, *Supplementary file 4*). We found significant enrichment of motifs known to be targets of Notch signaling within ±20 bp from the *Tfcp2l1* peak center, including Hes- and Tfap-, Atoh-, and Tcf-families. We confirmed that promoters of IC-specific genes bound *Tfcp2l1*, *Hes1* and *Foxi1* by using *Tfcp2l1* ChIP followed by *Hes1* or *Foxi1* ChIP (*Figure 8—figure supplement 1*). Considering that *Hes1* is a well-known repressor, these observations suggest a functional feedback loop from the Notch pathway to a subset of genes regulated by *Tfcp2l1*, potentially explaining why some IC genes are expressed at low levels in *Tfcp2l1*+*Hes1*+ PCs.

## *Jag1* is required for cellular patterning and differentiation of the collecting ducts

To analyze whether Notch signaling regulates the patterning of IC and PC proteins, we studied a conditional deletion of the Notch interacting domain exon 4 of Jag-1 (*Kiernan et al., 2006*). We deleted *Jag1* with either *Cdh16-Cre* or *Atp6b1-Cre*-drivers and the deletions were confirmed at P60 by immunostaining (*Figure 9—figure supplement 1*). *Cdh16-Cre;Jag1*f/f did not phenocopy *Cdh16-Cre;Tfcp2l1*f/f; in fact, ICs signature proteins were not only preserved in the *Jag1* knockout but were now widely co-expressed with PC proteins (e.g. *Tfcp2l1*+, *Atp6b1*+ with *Aqp2*+ or *Tfcp2l1*+, *Atp6b1*+ with *Krt8*+ or *Tfcp2l1*+, *Foxi1*+ with *Aqp2*+; *Figure 9*; *Figure 9—figure supplement 2*; n = 4 independent mice). Tubular structural failure was also found: tubules contained luminal debris (*Figure 9—figure supplement 2*) and collecting duct diameter was enlarged 28% in *Cdh16-Cre;Jag1*f/f (*Aqp4* staining facilitated the measurements: 30.1 ± 3.5 μm *vs* 23.5 ± 2 μm, SEM, n = 4 independent mice, p=$2\times10^{-14}$, *Figure 9—figure supplement 3*). *Krt8* was downregulated in these tubules, perhaps contributing to the structural failure. At later stages (P90) we observed hydronephrosis (3/4 mice), similar to another collecting duct-specific Notch-inactivation model (*Jeong et al., 2009*). These data suggest that while expression of IC proteins are 'upstream' of *Jag1*, their relative level in specific cells of the collecting duct is controlled by *Jag1* signaling.

To dissect the role of Jag-Notch in collecting cell development at higher resolution, we generated an IC *Atp6b1-Cre;Jag1*f/f;*mTmG* mouse and analyzed the *GFP*-labeled knockout cells. We found an increase in 'double positive cells' (from 2% to 12%, p=0.004) and a decrease in *Aqp2*+ PC cells (24% to 12%; p=$4.7E^{-06}$; n = 3 independent mice; *Figure 9—figure supplement 4*). Similar

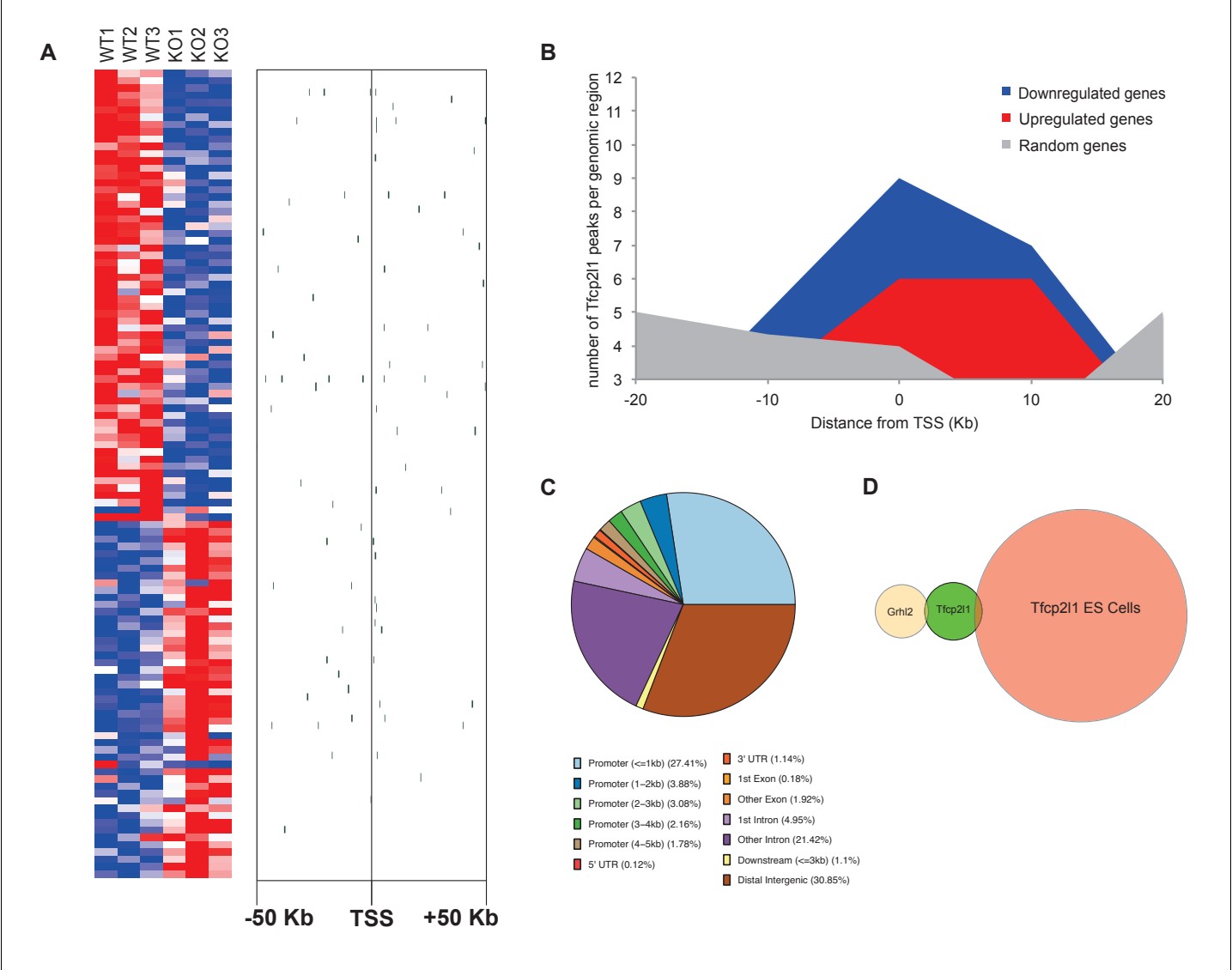

**Figure 5.** Identification of *Tfcp2l1* targets. (**A**) Identification of *Tfcp2l1* targets. Integration of knockout and *Tfcp2l1* ChIP-seq gene expression data obtained from P1 kidneys. Most of the genes significantly up or down regulated by *Tfcp2l1* (WT-*Tfcp2l1*^f/f vs KO-EIIACre;*Tfcp2l1*^f/f) demonstrated binding peak(s) mapping between +50 KB to −50 KB relative to the TSS for each gene. (**B**) *Tfcp2l1* peaks of both up and down regulated genes were enriched at the TSS in comparison with a random set of *Tfcp2l1* independent genes. (**C**) Genome wide annotation of *Tfcp2l1* peaks revealed that ~27% of peaks were within 1 kb of the TSS and 38% located within 10 kb from TSS. (**D**) Cell type specificity of *Tfcp2l1* ChIP peaks. Comparison of *Tfcp2l1* ChIP peaks in different models. P1 kidney (our study, Green) is compared with *Tfcp2l1* peaks identified in ES Cells (*Chen et al., 2008*, Red) and with *Grhl2* peaks identified in E18 kidney (*Werth et al., 2010*), Tan color. Note the limited overlap between these datasets.

The following figure supplements are available for figure 5:

**Figure supplement 1.** *Tfcp2l1* dependent genes localized to the collecting duct.

**Figure supplement 2.** Differentially expressed genes from kidneys of *Tfcp2l1* knockouts (*EIIA-Cre;Tfcp2l1*^f/f;>1.25 fold up- or down-regulated; p<0.05; n = 3 independent knockout and wild type mice).

data were obtained when we acutely interrupted Notch signaling by treating explanted E15 mouse kidneys with the gamma-secretase inhibitor Compound E (100 nM, 48 hr, n = 6 independent cultures of mouse kidneys). The treatment generated striking 'double positive' cells at the tips of the Ureteric

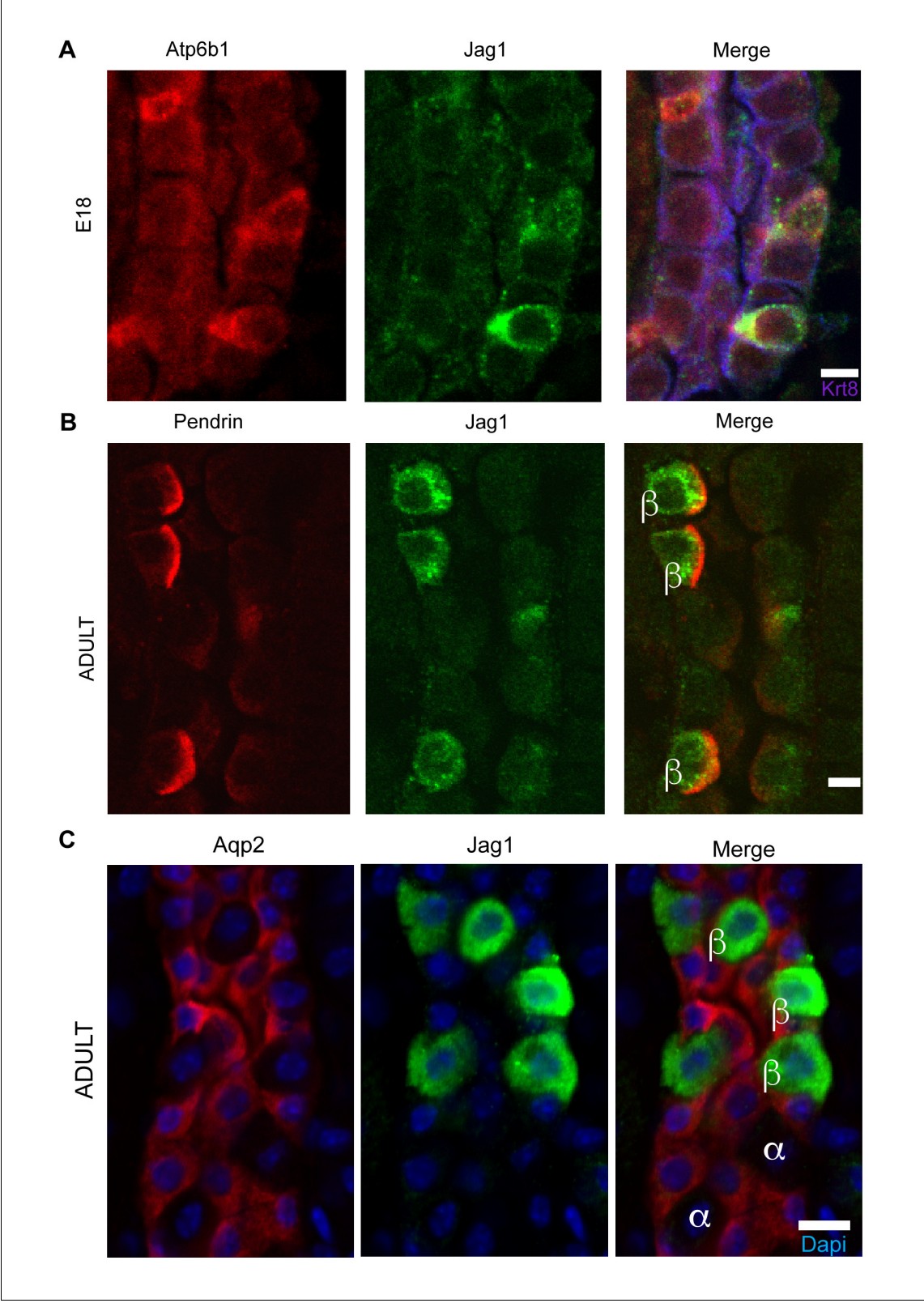

**Figure 6.** *Jag1* is a novel early marker of developing (E18) and adult (P60) IC cells and activated Notch is found in PCs  (A) *Jag1* (green) co-expressed with *Atp6b1* (red) at the first appearance of 'double positive' cells. *Krt8* (purple) is expressed by all cells at this stage. (B) In the adult kidney, *Jag1* (green) is specifically expressed in a subset of IC cells called Pendrin$^+$ $\beta$-ICs (red) (P60), but **C** not in other collecting duct cell types including *Aqp2$^+$* PC and *Aqp2$^-$* $\alpha$-ICs. Nuclei, blue (**A,B**) Bars = 5 μm. (**C**) Bar = 10 μm.

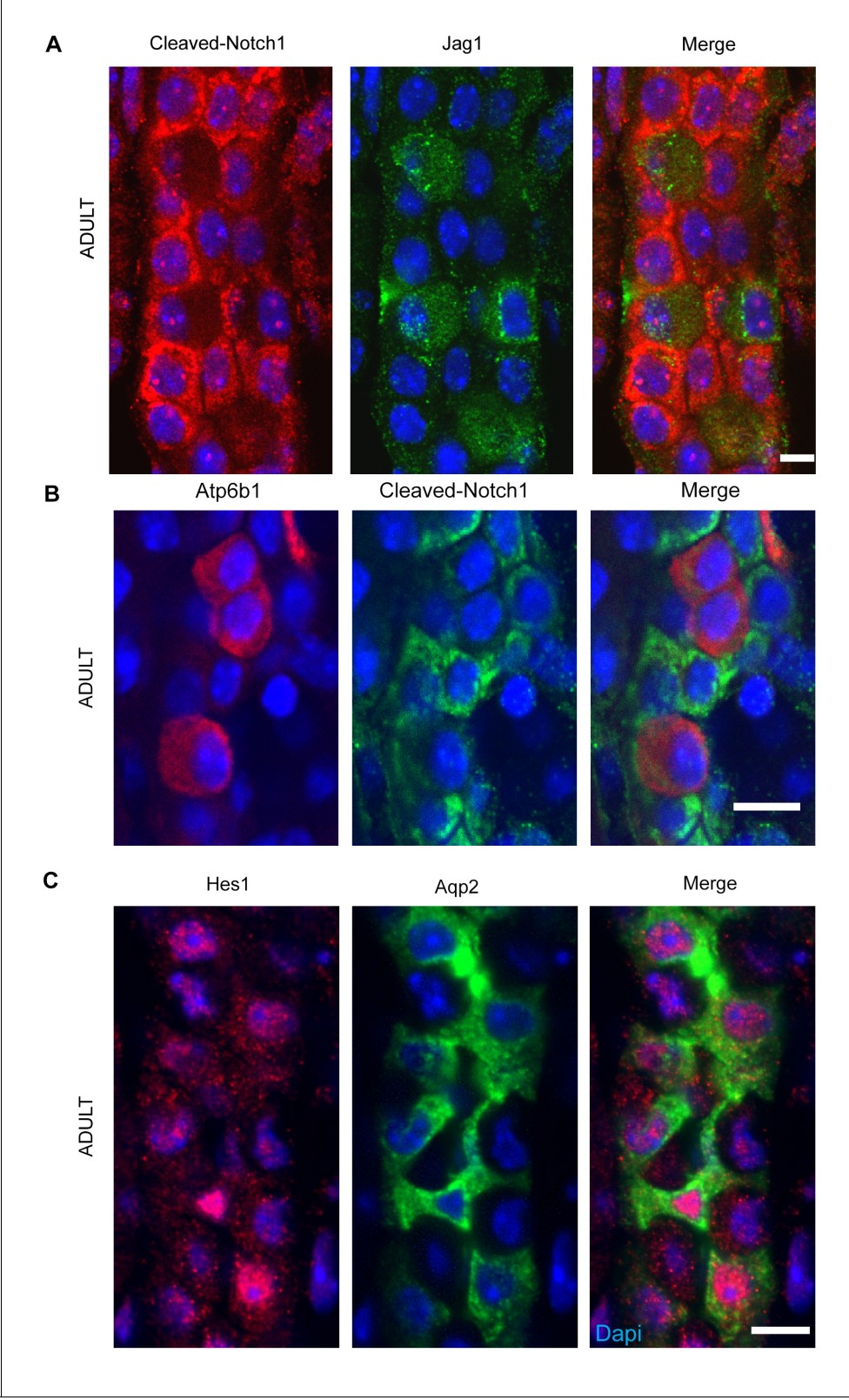

**Figure 7.** (A,B) *Jag1-Notch1* signaling in the collecting duct. Activated Notch is found in PC cells adjacent to *Jag1⁺Atp6b1⁺β*-ICs. (**C**) Consistently, Notch target gene *Hes1* (red) was detected in *Aqp2⁺* PC (green). **A, C** Bar = 10 μm, **B** Bar = 5 μm.

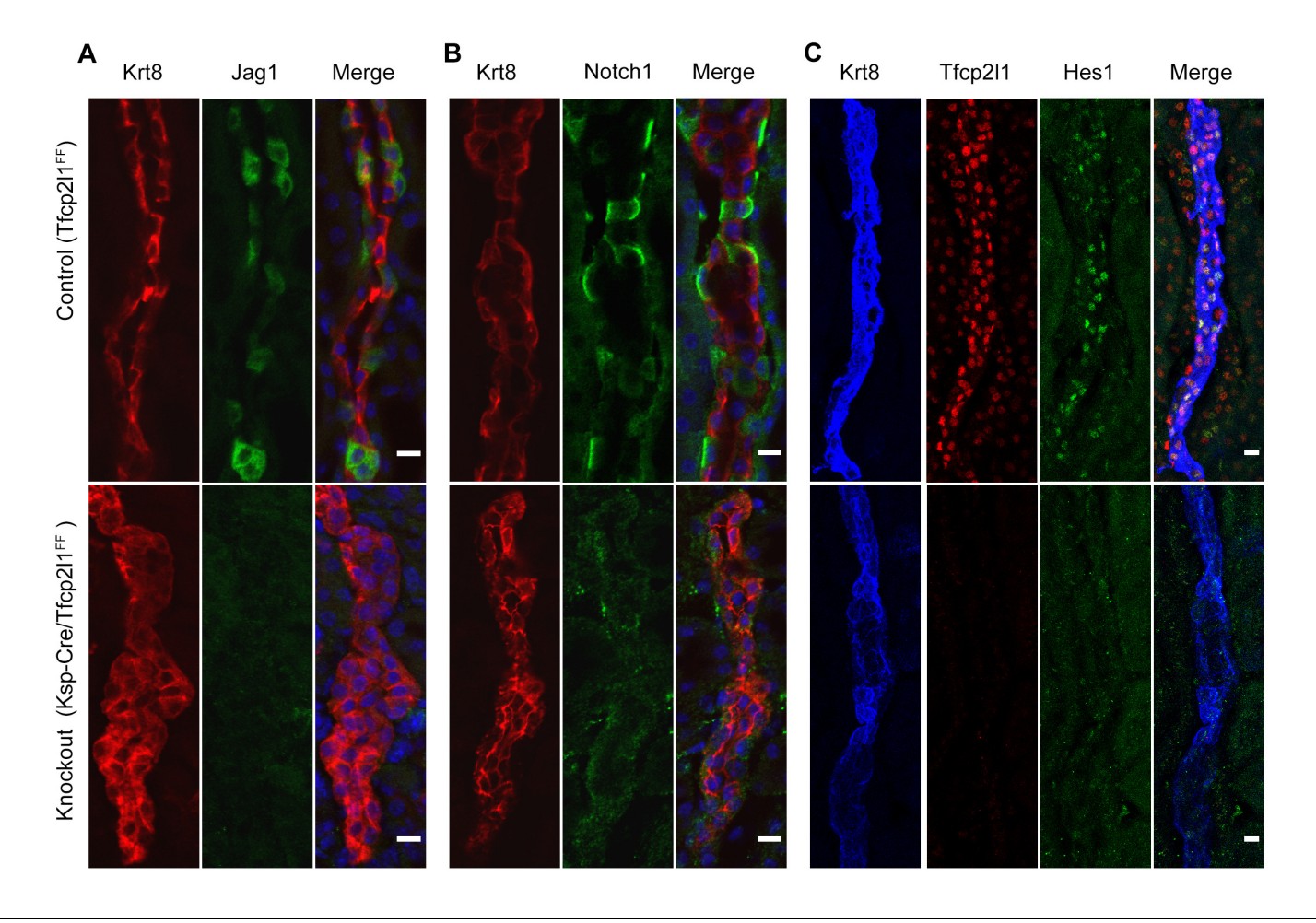

**Figure 8.** *Tfcp2l1* is required for Notch signaling in collecting ducts. *Tfcp2l1* knockout (*Cdh16-Cre;Tfcp2l1$^{f/f}$*) results in **A** depletion of *Jag1* (green) from *Krt8$^-$β*-ICs (red) **B**. depletion of cell surface (inactive) *Notch1* (green) from *Krt8$^-$ α*-ICs (red) **C** depletion of nuclear Hes (green) from *Krt8$^+$* PCs. (n = 4 independent mice for each immunodetection; Bars = 10 μm).

The following figure supplement is available for figure 8:

**Figure supplement 1.** Sequential Chromatin IP.

Bud (*Foxi1$^+$*, *Atp6b1$^+$* and *Krt8$^+$*; *Figure 10*). In sum, inactivation of *Jag1* disrupted IC and PC identity and arrested cells expressing inappropriate levels of IC and PC proteins.

## Discussion

While most segments of the kidney have one type of epithelia, multiple cell types are found in the collecting ducts including a diverse population of ICs, PCs and 'double-positive' progenitor cells. While these cells differ in structure, gene expression and function, they cooperate to perform physiologic functions. For example, Na$^+$ absorption by PCs via ENaC stimulates H$^+$ excretion by ICs, and conversely the blockade of Na$^+$ absorption in PCs by amiloride or by mutation of a subunit of the epithelial sodium channel results in the suppression of H$^+$ secretion creating an 'electrogenic' renal tubular acidosis (*Chang et al., 1996*; *DuBose and Caflisch, 1985*). Conversely, the deletion of bicarbonate transporters in ICs reduces the expression of Na+ channels in PCs (*Pech et al., 2010*). The physical interactions between ICs and PCs not only promotes functional coupling of these cells by

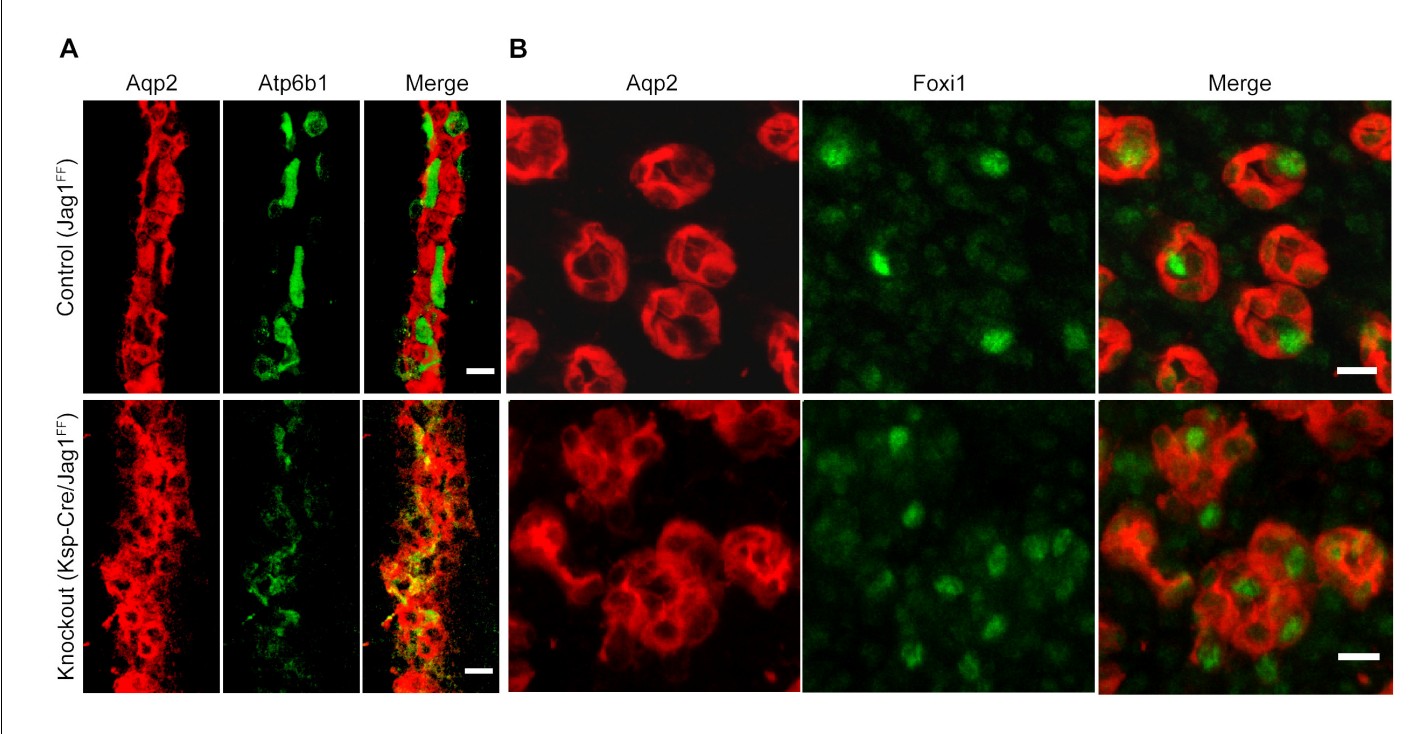

**Figure 9.** *Jag1* regulates the distribution of IC proteins in the collecting ducts. Knockout of *Jag1* (*Jag1^{f/f};Cdh16-Cre*) resulted in diffuse expression of IC proteins A *Atp6b1* and B *Foxi1*. Expression of these proteins overlapped with *Aqp2* creating 'double positive' cells. (n = 4 independent mice; Bars = 10 µm).

The following figure supplements are available for figure 9:

**Figure supplement 1.** Deletion of *Jag1* by *Cdh16-Cre*.

**Figure supplement 2.** Overview of the gross morphology of *Jag1* knockout kidneys.

**Figure supplement 3.** Jag1 is required for structural integrity of the collecting duct.

**Figure supplement 4.** Deletion of *Jag1* in IC (*Jag1^{f/f};Atp6b1-Cre;mTmG*) resulted in a six fold increase in 'double positive' cells (from 2% to 12%; n = 3 independent mice).

trans-epithelial gradients, but additionally permit juxtacrine signaling which regulates acid-base excretion (e.g. *CXCL12::CXCR4* (*Schwartz et al., 2015*).

ICs and PCs may serve opposing but linked functions such as K$^+$ excretion (PC) and K$^+$ absorption (IC) (*Park et al., 2012*; *Kim et al., 2016*). They may also serve independent roles which cannot be mutually compensated. For example, the decrease in PCs after Li treatment results in polyuria (*Christensen et al., 2004*) since PCs regulate water balance, while the decrease in ICs by *Tfcp2l1* deletion enhances kidney infection since ICs mediate urinary acidification and the production of a number of antimicrobials including *Lcn2*/NGAL (*Shohl and Janney, 1917*; *Paragas et al., 2014*). In sum, appropriate numbers of PCs and ICs are required for both physiologically linked and physiologically independent functions. Here we suggest that the integration of IC and PC populations is mediated by *Tfcp2l1* and *Jag1* and is reflected in their 'rosette' like architecture (e.g. *Figure 3*; *Figure 3—figure supplement 2B*).

*Tfcp2l1* and *Jag1* act on a primordial cell type which appears in the collecting duct between E15-E18 and expresses both IC and PC proteins (*Figure 2*; *Figure 2—figure supplements 1* and *2*). Descendants of these cells (tagged by *HoxB7-Cre* or *Atp6b1-Cre;mTmG*; *Figure 2B*) generated distinct ICs and PCs, yet even as canonical adult cells, they continued to express low levels of each

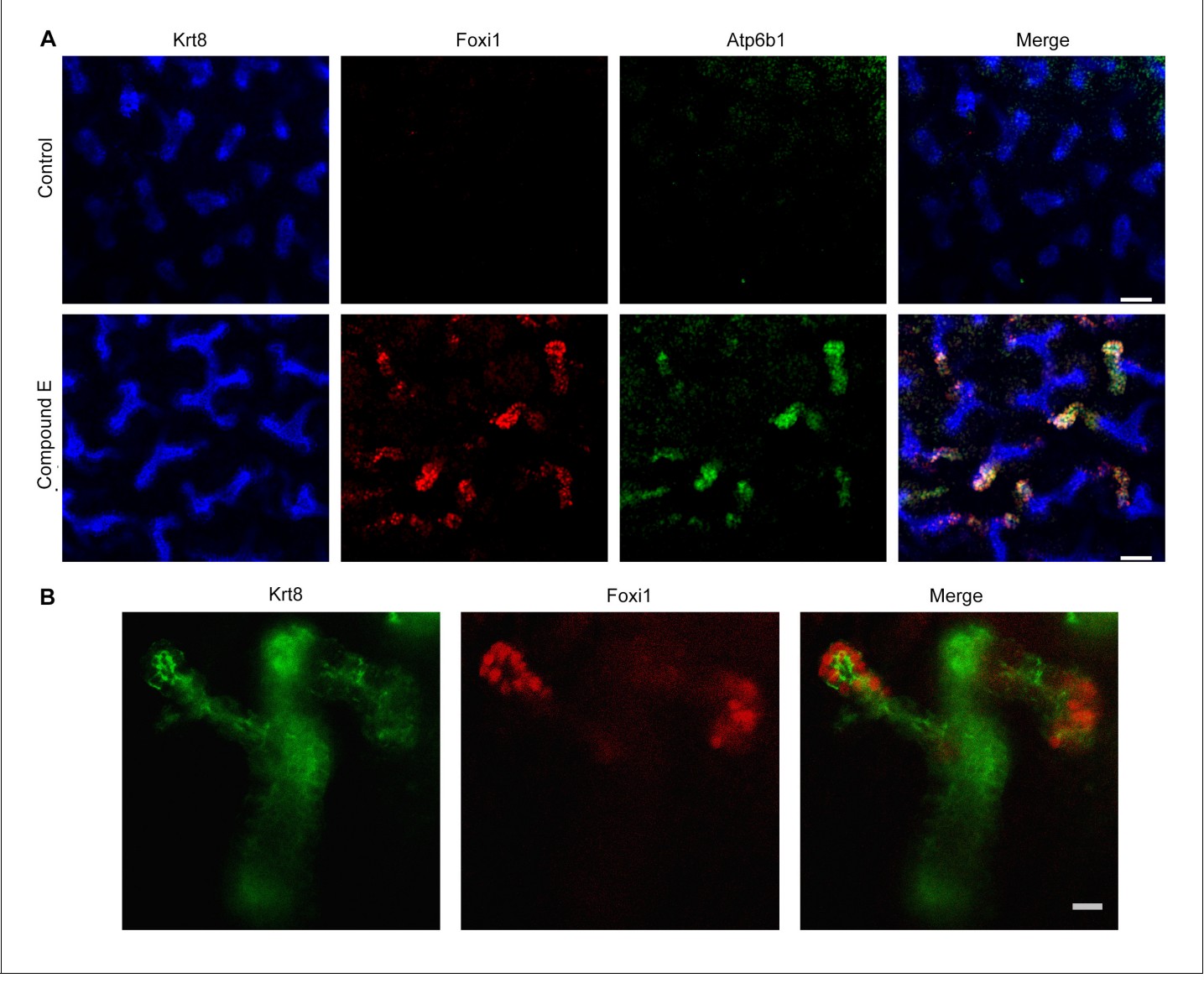

**Figure 10.** Manipulation of Notch signaling in vivo. (**A**) Inhibition of Notch signaling upregulates *Foxi1* and promotes IC cell differentiation. Acute inhibition of Notch signaling in E15 kidneys with Compound E (48 hr) resulted in the differentiation of IC cells at the tips of the UB/Collecting Ducts. Note that the IC cells demonstrated a 'double-positive' phenotype with the co-expression of PC (*Krt8*+ blue) and IC (*Foxi1*+ red, *Atp6b1*+ green) proteins. (**B**) High power of Compound E treated kidneys (n = 6 independent mouse kidney cultures). (**A**) Bars = 50 μm and (**B**) Bar = 10 μm.

other's signature proteins (*Figure 2—figure supplement 2*; *Figure 4B*), and remain plastic enough to change their phenotypes from PC to IC or IC to PC in vitro (*Fejes-Tóth and Náray-Fejes-Tóth, 1992*) and in vivo (*Park et al., 2012*; *Trepiccione et al., 2013*). In this setting the deletion of either *Tfcp2l1* (after it had already induced *Atp6b1*, *Figure 4*) or *Jag1* (*Figure 9*) demonstrated that cells initially destined to become ICs can fail to achieve their adult phenotype and demonstrate 'double positive' or even PC characteristics. In sum, the most likely explanation for our data is that collecting duct cells have the potential to choose either IC or PC fate and that the canonical cell types probably represent the extremes of a spectrum regulated by *Tfcp2l1* and *Jag1*.

*Tfcp2l1* appears to act on the IC-PC progenitors by both cell autonomous and cell non-autonomous mechanisms. Its cell autonomous activities were demonstrated both by an interaction of Tcfcp2l1 with IC specific genes (*Figure 5*; *Figure 5—figure supplements 1* and *2*) as well as loss of their expression in the *Tfcp2l1* knockout (*Figure 3*; *Figure 3—figure supplement 2*; *Figure 4*).

These cell autonomous activities may explain the absence of all stages of IC development in a cell population destined to express PC-like characteristics. These data are reminiscent of the deletion of *Foxi1* (*Blomqvist et al., 2004*) which lies downstream of *Tfcp2l1*.

The cell non-autonomous role of *Tfcp2l1* was evidenced by a direct interaction of *Tfcp2l1* and the *Jag1* promoter (quantified in Supplemental Table 3) and by the loss of *Jag1* expression in *Tfcp2l1* knockouts (*Figure 8*). Hence, *Tfcp2l1* induced *Jag1* which we propose mediates cell-non-autonomous signaling from *Jag1*+ICs to *Notch1*+ PCs.

Notch signaling is known to control cell fate decisions and to pattern many different organs through lateral-inhibition (*Afelik and Jensen, 2013*; *Martinez Arias et al., 2002*). In an initially homogeneous cell population, some cells start to express more Notch ligand which activates Notch expressing neighboring cells. Notch activation downregulates the ligand, which in turn leads to reduced Notch signaling in the ligand expressing neighbor, locking the two neighboring cells into a Notch OFF-state (ligand-expressing) or a Notch ON-state (Notch expressing). Notch activation initiates gamma-secretase-mediated cleavage of Notch-receptor permitting the Notch intracellular domain (NICD) to translocate to the nucleus where it acts as a transcriptional co-activator with RBPJ. NICD-RBPJ complex regulates a set of BHLH transcription factors including *Hey1*, *Hey2*, *Hes1*, *Hes5*.

Notch-mediated generation of multiple cell types has been previously identified in the zebrafish pronephric duct (*Liu et al., 2007*; *Ma and Jiang, 2007*) and in the Drosophila Malpighian tubule (*Wan et al., 2000*) which are both phylogenetically related to the collecting duct of the mammalian nephron, as well as in functionally related cells in the frog skin (*Quigley et al., 2011*). Many components of the Notch signaling pathway were expressed in the embryonic mammalian collecting duct. Before E18 there was faint expression of *Jag1* and *Notch1* throughout the collecting duct (data not shown), but by E18 there was clear excess expression of *Jag1* in a subset of duct cells which displayed IC and PC proteins. Surrounding these IC progenitors were PCs with activated nuclear *Hes1*, a pattern which became distinct after birth (*Figure 6*; *Figure 7*). In sum, the classical lateral-inhibition paradigm of Notch signaling appears to be relevant to patterning collecting duct cell types.

'Notch-ON' signaling may be relevant to cell fate choices because while *Hes1* targets are not known in detail, TF binding site overrepresentation analysis (PWMEnrich; *Thomas-Chollier et al., 2011*) of *Tfcp2l1* bound peaks demonstrated neighboring *Tfcp2l1* and *Hes1* binding sites, as well as combinatorial binding of *Tfcp2l1-Foxi1-Hes1* (*Figure 8—figure supplement 1*) raising the possibility that *Hes1* signaling in PCs may negatively modulate *Tfcp2l1* and *Foxi1* targets. *Hes1* might repress IC genes in PC cells by enhancing chromatin modification enzymes since the inactivation of histone methyltransferase Dot1l appeared to convert PCs into ICs (*Xiao et al., 2016*). In sum, Notch signaling might antagonize *Tfcp2l1* in PC cells where Notch activates *Hes1*.

'Notch-OFF' in IC cells may result from the basolateral localization of Notch (perhaps making it sterically inaccessible to ligands), or perhaps as a result of the expression of NUMB in some ICs (not shown; *McGill et al., 2009*). The 'Notch-OFF' state is likely to be critical for IC function because it may link *Tfcp2l1* to *Foxi1*. While *Tfcp2l1* might stimulate the expression of *Foxi1* directly (*Tfcp2l1* bound *Foxi1* at +50 kb from the TSS), the upregulation of *Foxi1* appeared to be a direct consequence of Notch-OFF because activated Notch signaling negatively regulated *Foxi1* (*Guo et al., 2015*) and conversely Notch inactivation models (*Jag1* knockout and pharmacologic inhibition) generated diffuse expression of *Foxi1* (*Figure 9*, *Figure 10*). In addition, we found that *Foxi1* bound to the *Jag1* promoter (*Figure 8—figure supplement 1*), consistent with observations made in the endolymphatic duct cells of *Foxi1* knockouts where *Jag1* was absent (*Hulander et al., 2003*). Hence, while *Tfcp2l1* and *Foxi1* may act independently and synergistically to upregulate the IC phenotype (*Blomqvist et al., 2004*; *Vidarsson et al., 2009*), we expect that a positive regulatory loop between *Foxi1* and *Jag1* maintains high levels of *Jag1* expression in ICs, but conversely inhibits *Foxi1* in neighboring PCs. Hence, the induction of *Jag1* by *Tfcp2l1* suppressed IC gene expression in PCs.

Our data provide a developmental explanation for experiments demonstrating that the balance of cell types in the adult collecting duct can be modified by Notch signaling. For example, conditional inactivation of *Mib1* (*Jeong et al., 2009*; *Nam et al., 2015*), a E3 ubiquitin-ligase that positively regulates Notch signaling by regulating Notch ligands, resulted in more ICs as did the inactivation of a cell surface protease, *Adam10* (*Guo et al., 2015*), which regulates the cleavage and activation of Notch proteins. Conversely, overexpression of Notch intracellular domain ICD directed collecting duct cells towards a PC fate (*Jeong et al., 2009*). While these data were observed in the adult, we suggest that each of these experimental manipulations may have acted at the embryonic

IC-PC 'double positive' stage. In other words, rather than a binary choices between canonical IC and PC phenotypes, a large body of literature may be explained by the notion that the manipulation of Notch signaling caused a failure to choose among competing programs of gene expression, producing the 'double positive' phenotype.

Our model has subtle differences with classical demonstrations of Notch signaling. In other developing organs, including the inner ear and intestine, classical negative feedback loops act in a paracrine manner between neighboring cells to inhibit the genes which induced the Notch ligands (e.g. *Ngn1* induces expression of *Dll1*, but Notch activation inhibits *Ngn1*; *Atoh1* activates *Dll1/4*, but *Dll1/4* suppresses *Atoh1* [*Morrison et al., 1999*; *Kazanjian et al., 2010*]). In contrast, *Tfcp2l1* expression was inhibited only by two fold in *Hes1*⁺ PCs (IC>PC). While this differential expression might contribute to cell speciation by driving higher levels of *Jag1* expression in ICs, we note that *Foxi1*, which is downstream of *Tfcp2l1*, is not only critical for *Jag1* expression, but it is reciprocally inhibited by Notch signaling (*Figures 9* and *10*). Consequently, both *Tfcp2l1* and *Foxi1*, rather than either alone, may mediate the Notch signaling.

A second distinction with the classical models was the response of PCs to the loss of *Hes1*. In other developing organs, loss of *Hes1* signaling resulted in a failure of tissue development (e.g. spinous skin cells; *Blanpain et al., 2006*). In contrast, *Tfcp2l1* deletion and the failure of *Hes1* signaling did not block PC development, indicating that *Hes1* may play an inhibitory role in the collecting duct by preventing IC development in cells otherwise competent to become PCs. Rather than *Hes1*, additional Notch targets such as *Elf5* may mediate Notch dependent PC development (*Grassmeyer et al., 2017*).

The plasticity of collecting duct cells was identified in the adult (*Schwartz et al., 1985*); here we show that this plasticity reflects a developmental program that may remain active long after birth and is likely to permit adaptations to environmental cues. *Tfcp2l1* was found in a screen for transcription factors which induce epithelialization (*Barasch et al., 1999*; *Schmidt-Ott et al., 2007*); Werth and Leete, Unpublished). However its key role lays in regulating the plasticity of collecting duct progenitors (*HoxB7*⁺, *Atp6b1*⁺, *Jag1*⁺ cells; Model *Figure 11*). *Tfcp2l1* provides essential directions for the initiation and refinement of this cell type via cell autonomous and non-cell autonomous mechanisms. As a consequence, it regulates the cellular diversity, which is the principal feature that distinguishes the collecting duct from other segments of the nephron.

## Materials and methods

*Animal models Tfcp2l1* conditional knockouts were generated by flanking exons 3, 4 and 5 (~1.8 kb) with LoxP sites. These exons included the CP2 conserved functional domain and their deletion created an open reading frame shift. The targeting construct was generated by using a modified NCI BAC recombineering protocol. Briefly, mouse (C57) *Tfcp2l1* BAC clone (RP24-291G6, CHORI, Oakland, California) was recombineered using Loxp-Neomycin-Loxp (LNL) cassette with homologous arms upstream of exon 3 and Frt-Neomycin-Frt-Loxp (FNFL) cassette downstream of exon5. The *Tfcp2l1* -targeting BAC DNA was electroporated into mouse KV1 ES cells (generated by Columbia University Transgenic Facility), followed by neomycin selection. Neomycin-resistant ES clones were screened using PCR and Southern Blots and selected clones were microinjected into C57B6 blastocysts to generate chimeras. The *Tfcp2l1* chimeric mice were subsequently mated with *EIIa-flpe* C57B6 female mice for both germline transmission and flpe-based removal of the Frt-floxed neomycin cassette, which generated F1 *Tfcp2l1*-floxed mice. Cre mediated deletion was confirmed at the genomic level using PCR, at the RNA level using quantitative Real-Time PCR and at protein level using immunohistochemistry. *Tfcp2l1* knockout and wild type kidneys were assayed using Affymetrix Mouse Genome 430.2 microarrays.

*EIIA-Cre* (RRID:IMSR_JAX:003724), *Cdh16-Cre* (RRID:IMSR_JAX:012237) and *Jag1*^f/f (RRID:IMSR_JAX:010618) were purchased from the Jackson Labs, Rosa26*mTmG* (RRID:IMSR_JAX:007576) was kindly provided by C Mendelsohn and *Atp6b1-Cre* was a gift from R Nelson, Utah. Rosa26*mTmG* genetic reporters express membrane-bound green fluorescein protein (*GFP*) upon Cre-mediated recombination (*Muzumdar et al., 2007*). *GFP* was detected with anti-*GFP* post-antigen retrieval. All rodents experiments were approved by Columbia University IACUC.

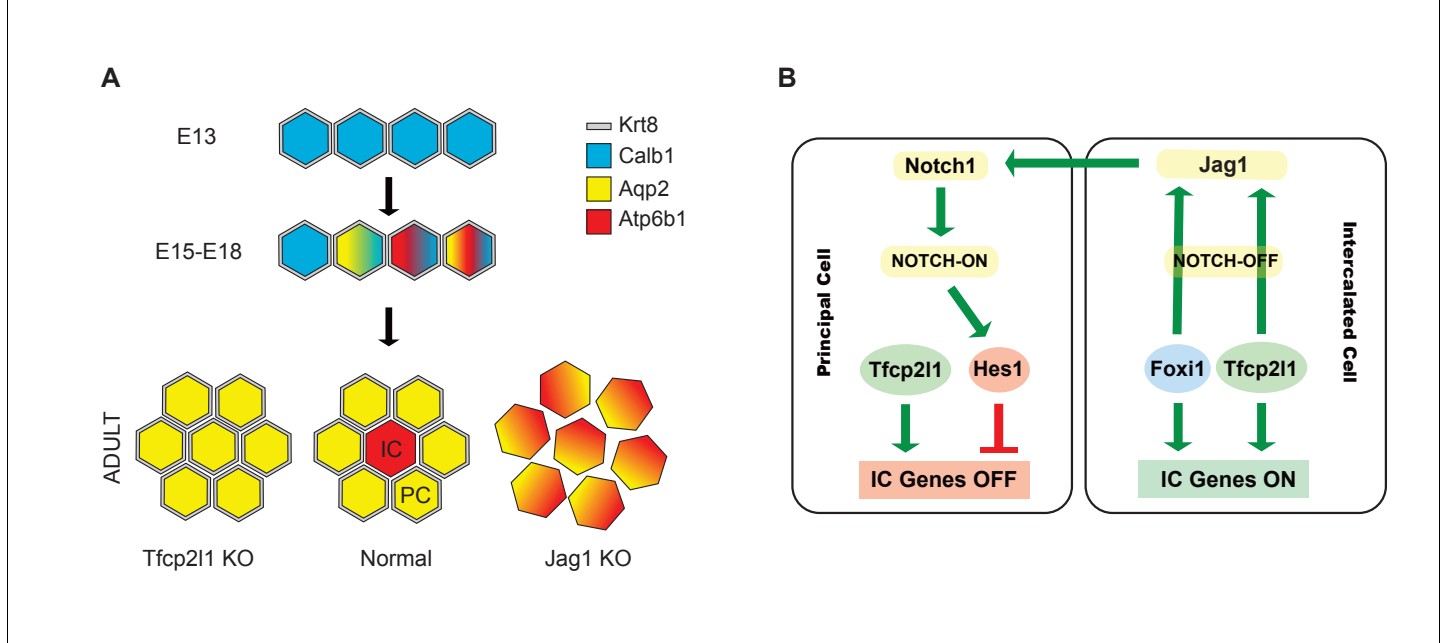

**Figure 11.** Models. (A) Development of cellular diversity in the collecting duct. Initially, we found a monotonous expression of PC proteins at E13, and then a transitional stage (E15–E18) characterized by the appearance of 'double-positive' cells. After birth, these cells achieved distinct identities and assumed rosette-like patterning. Deletion of *Tfcp2l1* resulted in a monotonic cell type expressing PC cell, but not IC cell markers. Inactivation of *Jag1* in contrast resulted in the loss of cell identity and patterning, increasing the number of 'double-positive' cells typified by E18 collecting ducts. (B) Proposed model of cell-autonomous and non-cell-autonomous actions of *Tfcp2l1*. *Tfcp2l1* induces the expression of IC genes, including *Jag1*. Expression of *Jag1* triggers Notch signaling in adjacent PC cells. *Jag1* signaling results in activation of *Hes1* (Notch-ON) in PCs and conversely *Foxi1* expression in ICs (Notch-OFF). We envision that the combination of *Tfcp2l1* with either *Foxi1* or *Hes1* drives cell identity. Maintenance of this circuit may depend on an excess of *Tfcp2l1* in ICs (*Figure 1*) as well as the expression of *Foxi1* which is known to induce *Jag1*. Conversely, *Jag1* suppression of *Foxi1* (*Figures 8* and *9*) demonstrates a negative feedback loop in neighboring PCs.

## Chromatin immunoprecipitation

As described (*Werth et al., 2010*), P1 mouse kidneys were cut into pieces and then sieved to single cells in DMEM. Chromatin was crosslinked and fragmented for 15 min (30 s stroke/30 s pause) using a Fisher Sonicator 450 (power 8; on 10 s and off 20 s for 6 min) to a fragment size of 100–500 bp. DNA concentration was measured by NanoDrop and then subjected to *Tfcp2l1* ChIP with anti-human *Tfcp2l1* (RRID:AB_2202564) or goat IgG control (Jackson Immunoresearch). The ChIP antibody was extensively validated in ChIP-PCR assays. In brief, we used conservative *Tfcp2l1* binding site at Krt7 promoter to validate the performance of the *Tfcp2l1* antibody in a ChIP PCR assay. A ChIP Library was prepared according to the Illumina ChIPseq library preparation kit using size selection (150–250 bp). The library was sequenced on Illumina Genome Analyzer IIx (San Diego, California). Sequences were aligned to mm9 genome assembly using Bowtie (*Langmead et al., 2009*) and binding sites were detected using MACS (*Zhang et al., 2008*). We used GREAT (*McLean et al., 2010*) for gene association/enrichment analysis. For motif analysis we used PWMEnrich (http://www.bioconductor.org/packages/release/bioc/html/PWMEnrich.html)

## RNA extraction and cDNA synthesis, Affymetrix Arrays

We isolated RNA using Qiagen RNAeasy micro kit, analyzed the isolate by Bioanalyzer (Agilent, Santa Clara, California), and synthesized cDNA using kits from Applied Biosystems. Preparation of templates and hybridization on Affymetrix Arrays was performed at the Columbia Genome Center. Differential gene expression was identified using R Limma (https://bioconductor.org/packages/limma). Induced genes were validated by QPCR using Sybr Green (Applied Biosystems, Forster City, California) and relative levels of mRNA expression were normalized to *β*-actin mRNA. Primer pairs were designed using Primer3 software (Whitehead Institute, MIT) and validated by product size

(*Supplementary file 5*). P-values cut-offs (0.05 and 0.01) were chosen to generate datasets of comparable sizes.

Our data has been deposited with GEO, Accession Number GSE87769 (and Super Series: Accession Number GSE85325, GSE87744, GSE87752). *Tfcp2l1* ChIP-Seq data from ES cells was obtained from GEO, Accession Number GSE11431. *Grhl2* ChIP-Seq data from the kidney was obtained from GEO, Accession Number GSE24295.

### In situ hybridization and immunocytochemistry

Tissues were prepared by intracardial perfusion with PBS followed by 1% PFA/PBS, 30% Sucrose/PBS, OCT (Tissue-Tek) and cryosectioned at 50 μm. Blocking and staining was performed in PBS/3% BSA/0.2% Triton X-100 solution. Controls and experimental samples were always placed on the same slide, stained in parallel and imaged with the same parameters using confocal Zeiss LSM510 Confocal Microscope (Columbia Core Facilities). We used confocal microscopy to reconstruct whole cells from the image stacks and estimated the amount of protein per cell by measuring total fluorescent signal. Antibodies are listed in *Supplementary file 5*. Data were analyzed using Fiji (ImageJ). Digoxigenin-labeled antisense riboprobes were generated from cDNAs using reverse primers containing a 5' T7 polymerase promoter sequence (*Supplementary file 5*; *Schmidt-Ott et al., 2005*).

## Additional information

### Funding

| Funder | Grant reference number | Author |
| --- | --- | --- |
| Deutsche Forschungsgemeinschaft | FOR1368 | Kai M Schmidt-Ott |
| Deutsche Forschungsgemeinschaft | FOR667 | Kai M Schmidt-Ott |
| Urological Research Foundation Berlin | | Kai M Schmidt-Ott |
| Deutsche Forschungsgemeinschaft | Emmy Noether | Kai M Schmidt-Ott |
| National Institutes of Health | RO1DK073462 | Jonathan Barasch |
| March of Dimes Foundation | Research Grant | Jonathan Barasch |
| National Institutes of Health | RO1DK092684 | Jonathan Barasch |
| National Institutes of Health | U54DK104309 | Jonathan Barasch |

The funders had no role in study design, data collection and interpretation, or the decision to submit the work for publication.

### Author contributions

MW, AQ, Formal analysis, Investigation, Methodology; KMS-O, Conceptualization, Formal analysis, Writing—original draft, Writing—review and editing; TL, Data curation, Formal analysis, Investigation, Methodology; CH, Data curation, Formal analysis; MV, Data curation, Investigation; NP, WY, PL, XC, AS, Formal analysis; CJS, QA-A, Writing—review and editing; WM, AR, Formal analysis, Investigation; C-SL, Resources, Investigation, Methodology; JK, Writing—original draft, Writing—review and editing; JB, Conceptualization, Writing—original draft, Writing—review and editing

### Author ORCIDs

Max Werth, http://orcid.org/0000-0003-0169-6233
Qais Al-Awqati, http://orcid.org/0000-0001-7141-1040
Jonathan Barasch, http://orcid.org/0000-0002-6723-9548

### Ethics

Animal experimentation: All experiments were approved by the Institutional Animal Care and Use Committee (IACUC) at Columbia. Protocol # AC-AAAH7404.

# Additional files

## Supplementary files

• Supplementary file 1. Limma analysis of Affymetrix microarrays demonstrating differentially expressed genes in *EIIA-Cre;Tfcp2l1*$^{f/f}$ vs *Tfcp2l1*$^{f/f}$ kidneys (P1).

• Supplementary file 2. Identification of *Tfcp2l1* binding sites in P1 kidney using ChIP-seq (compared to IgG ChIP). Analysis by MACS.

• Supplementary file 3. GREAT analysis of *Tfcp2l1* binding sites from *Supplementary file 2* demonstrating significantly enriched pathways.

• Supplementary file 4. Motif analysis of top 1000 *Tfcp2l1* binding sites from P1 kidney (data from *Supplementary file 2*).

• Supplementary file 5. PCR primers and antibodies.

## Major datasets

The following datasets were generated:

| Author(s) | Year | Dataset title | Dataset URL | Database, license, and accessibility information |
|---|---|---|---|---|
| Werth et al. | 2017 | Identification of Tfcp2l1 target genes in the mouse kidney | https://www.ncbi.nlm.nih.gov/geo/query/acc.cgi?acc=GSE87769 | Publicly available at the NCBI Gene Expression Omnibus (accession no: GSE87769) |
| Werth M, Barasch J | 2017 | Tfcp2l1 controls cellular patterning of the collecting duct. | https://www.ncbi.nlm.nih.gov/geo/query/acc.cgi?acc=GSE85325 | Publicly available at the NCBI Gene Expression Omnibus (accession no: GSE85325) |
| Werth M, Barasch J | 2017 | RNA-Seq Gene Expression Analysis in Rat Metanephric Mesenchyme overexpressing Transcription Factor Tfcp2l1 | https://www.ncbi.nlm.nih.gov/geo/query/acc.cgi?acc=GSE87744 | Publicly available at the NCBI Gene Expression Omnibus (accession no: GSE87744) |
| Werth M, Barasch J | 2017 | Genome wide map of Tfcp2l1 binding sites from mouse kidney | https://www.ncbi.nlm.nih.gov/geo/query/acc.cgi?acc=GSE87752 | Publicly available at the NCBI Gene Expression Omnibus (accession no: GSE87752) |

The following previously published datasets were used:

| Author(s) | Year | Dataset title | Dataset URL | Database, license, and accessibility information |
|---|---|---|---|---|
| Wei C | 2008 | Mapping of transcription factor binding sites in mouse embryonic stem cells | https://www.ncbi.nlm.nih.gov/geo/query/acc.cgi?acc=GSE11431 | Publicly available at the NCBI Gene Expression Omnibus (accession no: GSE11431) |
| Schmidt- Ott KM, Barasch J, Walentin K, Werth M | 2010 | Gene expression in epithelial and non-epithelial cells of renal origin | https://www.ncbi.nlm.nih.gov/geo/query/acc.cgi?acc=GSE24295 | Publicly available at the NCBI Gene Expression Omnibus (accession no: |

GSE24295)

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
