## [Decision Letter]

Thank you for submitting your article "*TFCP2L1* Regulates the Patterning of Cells in the Collecting Duct of the Kidney" for consideration by *eLife*. Your article has been reviewed by three peer reviewers, and the evaluation has been overseen by a Reviewing Editor and Harry Dietz as the Senior Editor. The reviewers have opted to remain anonymous.

The reviewers have discussed the reviews with one another and the Reviewing Editor has drafted this decision to help you prepare a revised submission.

Summary:

This is a detailed analysis of the Tfcp2l1 knockout in the distal nephron. The authors reveal that this gene is necessary for promoting the intercalated cell phenotype within collecting ducts cells that are capable to be either principal or intercalated cells, at least partially through regulation of the Notch signaling pathway. These conclusions are drawn based on in situ antibody staining, RNA-Seq analysis and ChIP-seq. The authors devise a model whereby Tfcp21l interacts with Notch signaling to repress the intercalated cell fate in plastic collecting duct cells. This paper makes a number of important observations and establishes a new paradigm for collecting duct differentiation.

Essential revisions:

The work was found to be potentially interesting and worthy of publication in *eLife*, however there were major concerns from all three of the reviewers, all of which need to be addressed. They are:

1) The writing needs to be polished. The data presentation is so dense that one cannot determine whether the conclusion is reasonable or not. There are innumerable small images presented to support their conclusions, but many of them are over processed, and display a confusing array of overlapping colors (although some individual wavelengths are shown in supplementary data, which is necessary), and from poorly defined tubule segments (in terms of their overall location in the kidney), that it is difficult to assess the overwhelming amount of data from the different animals and in vitro models used here. The overall major problem of information overload and seeming over interpretation of some of the images needs to be addressed. This will require serious editing and the removal of all but the essential data that support the hypothesis that Tfc2l1 is involved here.

2) The authors might want to address in their (already too lengthy) Discussion, physiological contexts in which the cellular composition of the collecting duct has been shown to change. For example, with lithium exposure.

3) Citations to figures in the text are incorrect. Some panels are never referred to. Some data is presented but the location of that data is not mentioned (Figure 4 is particularly bad). A lot of data is mentioned but there is no corresponding figure or supplemental figure. This needs to be corrected.

4) A second issue is the experiments regarding the transfection of isolated MM with Tfcp2l1. This line of experimentation was very confusing to me until I read the Materials and methods section in which it was indicated that this was actually how this factor was discovered. However, that is not mentioned in the Results section. The reason this is confusing is because the manuscript focuses on the collecting ducts which are derived from a different lineage from the MM. Why they would use the MM instead of a collecting duct cell line is confusing. Further it is confusing as to how they are able to get cell types from both the MM lineage as well as the collecting duct lineage from this experiment. It appears as if they get an entire patterned nephron and collecting duct. Is Tfcp2l1 expressed in all these cells? Is this result different than if they treated with LIF? In the experiments where they compare the Tfcp2l1 transfected to GFP, do they treat with LIF (not clear from the Methods). If not, what are they comparing? An entire nephron to uninduced MM? If so, what is the relevance of these results? Without better explanation and rationale, I feel these results should be excluded from this paper.

5) Some of the data if very difficult to see (Figure 1P-R, Figure 2G-J, 3C, 4A, B, most of Figure 6). In some cases, individual channels need to be shown. In others, higher magnification images would be of help.

6) The authors need to clarify what Notch antibody they are using in their stainings. There are discrepancies between what they say in the text, how the label their figures and what the figure legends say.

7) Figure 6A and B need to be divided up into distinct panels as each panel shows a different antibody combination from a different kidney (or at least region of the kidney). The single channel staining of pendrin is of low quality and hard to interpret. Co-stainings and/or higher mag should be shown.

8) Figure 1, nuclear Tfcp staining is clear in the adults (E, F) but less so in the embryos (B, C, D). Do the authors have any better images? Also, is it clear that Tfcb is in all collecting duct cells in that levels are equal? Similarly, in 1M, the triple staining makes it hard to see whether Tfcb is in all cells. In 2E, it looks like the IC cells have more Tfcb for example.

9) In Figure 3 the authors never show us the expression of Tfcp in the Adeno infected cultures, so it is really hard here to discern how much of the epithelia is due to Tfcb expression and how much is due to the supposed induction by LIF. Again, the multiple stains make it really hard to discern individual patterns. In 3D, it is hard for any reviewer to discern whether the Tfcb ChIP-seq peaks are meaningful and whether they localize to specific regions of genes or at the TSS. From the excel sheet, individual peaks may be significant but can the authors say something about the statistical significance of location of peaks? In other words, are peaks more likely to be near TSS rather than dispersed more randomly at target genes?

10) In Figure 6, it would be nice to see Tfcb expression in the Jag1 KOs. In other words, does notch signaling alter levels of Tfcb and does this in turn reinforce more notch signaling to set boundaries between cell types?

11) The model in Figure 7 assumes equal amounts of Tfcb, but this not clear. The transition from double positive to single positives, clearly require notch/jag1 and this depends on Tfcb, which is expressed in all cells (I think). But as notch signaling commences, do Tfcb levels remain the same or are they regulated by notch and does this then alter the amount of notch ligands to reinforce the pattern? The model assumes Hes1 and Foxi1 mediate the activation and repression, but again I am not sure Tfcb levels are equivalent in the two cell types.

---

## [Author Response]

*Essential revisions:*

*The work was found to be potentially interesting and worthy of publication in eLife, however there were major concerns from all three of the reviewers, all of which need to be addressed. They are:*

*1) The writing needs to be polished. The data presentation is so dense that one cannot determine whether the conclusion is reasonable or not. There are innumerable small images presented to support their conclusions, but many of them are over processed, and display a confusing array of overlapping colors (although some individual wavelengths are shown in supplementary data, which is necessary), and from poorly defined tubule segments (in terms of their overall location in the kidney), that it is difficult to assess the overwhelming amount of data from the different animals and in vitro models used here. The overall major problem of information overload and seeming over interpretation of some of the images needs to be addressed. This will require serious editing and the removal of all but the essential data that support the hypothesis that Tfc2l1 is involved here.*

We appreciate the comments and have spent much effort reorganizing a number of images, displaying separate confocal channels, removing (or moving to supplemental images) all but the most essential data rather than document all of the IC and PC proteins at different stages in a single image, and we describe the location of the images (cortical or medullary collecting ducts). In all we have removed 20 panels but at the same time we are now showing individual channels. In particular, E15 data is no longer shown because of very rare presence of double positive cells before E18.

We should mention that because we have reduced the complexity of the images, there are instances where we did have to indicate “data not shown”. We are happy to replace those images-perhaps in other types of files – according to your guidance.

Next, we simplified the writing of the paper along with clarification of the data presentation, including removing the section on induction in vitro, the section on additional segments of the nephron, especially the connecting segment, and revisions to reduce redundant statements. As a result, we reduced the manuscript by 500 words in order to focus on the Tfcp2l1 pathway.

*2) The authors might want to address in their (already too lengthy) Discussion, physiological contexts in which the cellular composition of the collecting duct has been shown to change. For example, with lithium exposure.*

We have expanded the section in the Discussion concerning medical settings in which IC and PC ratios might change.

In addition, these studies are ongoing in our lab but the connection between different diseases of collecting ducts (obstruction, UTI, Renal Tubular Acidoses, Li exposure) and the underlying signaling pathways remains unexplored. We suspect a variety of signaling paradigms are recruited in different medical settings. In Figure 12 we can see that PC cells have activated Wnt signaling (green), which most likely plays a role in Li exposure.

Author response image 1.**DOI:**
http://dx.doi.org/10.7554/eLife.24265.030

Consequently, we comment on the few known physiological examples of perturbation of the IC-PC ratio, but there is much unknown.

*3) Citations to figures in the text are incorrect. Some panels are never referred to. Some data is presented but the location of that data is not mentioned (Figure 4 is particularly bad). A lot of data is mentioned but there is no corresponding figure or supplemental figure. This needs to be corrected.*

We have reformatted all of the images and we have carefully paired the text and the figures.

In order to cut down the amount of data (comment #1), we had to remove some of the figures particularly the evidence of very rare IC cells at E15, multiple combinations of IC and PC marker proteins, and demonstrations of loss of protein expression in knockouts. Instead we have indicated “not shown”. We are happy to replace any additional data as per the Editor’s recommendation.

*4) A second issue is the experiments regarding the transfection of isolated MM with Tfcp2l1. This line of experimentation was very confusing to me until I read the Materials and methods section in which it was indicated that this was actually how this factor was discovered. However, that is not mentioned in the Results section. The reason this is confusing is because the manuscript focuses on the collecting ducts which are derived from a different lineage from the MM. Why they would use the MM instead of a collecting duct cell line is confusing. Further it is confusing as to how they are able to get cell types from both the MM lineage as well as the collecting duct lineage from this experiment. It appears as if they get an entire patterned nephron and collecting duct. Is Tfcp2l1 expressed in all these cells? Is this result different than if they treated with LIF? In the experiments where they compare the Tfcp2l1 transfected to GFP, do they treat with LIF (not clear from the Methods). If not, what are they comparing? An entire nephron to uninduced MM? If so, what is the relevance of these results? Without better explanation and rationale, I feel these results should be excluded from this paper.*

Thank you for these comments. The reviewer is correct that we identified Tfcp2l1 from an induction screen – that is, we found that a single transcription factor can induce an entire patterned nephron and collecting duct. In fact, Tfcp2l1 is active without the addition of any other growth factors to metanephric mesenchymal cultures (LIF is not needed), and its induction rate approached 100% using two different adenoviral preparations. In the absence of growth factors, metanephric mesenchyme survives for one-two days, but after Tfcp2l1 induction, the mesenchyme expands and segmented nephrons appear. In our experiments, Tfcp2l1 was the only transcription transferred by Adenovirus to the mesenchyme that could reproduce LIF activity in rat metanephric mesenchyme.

Tfcp2l1 is expressed weakly in proximal segments of the nephron, more strongly in the Talh and most strongly and persistently in the collecting ducts. This may be an explanation for its remarkable inductive activity of both proximal and distal nephron segments. We think that the readout not only reflected the earlier expression of Tfcp2l1 throughout the nephron, but may reflect that both the ureteric bud and the metanephric mesenchyme derive from the intermediate mesoderm.

However, we appreciate that the induction of many types of epithelia including ureteric epithelia is not a classical finding in our field, and hence requires more explanation which is as the reviewer indicated is really distinct from the main findings of the paper. In addition, because our Tfcp2l1 knockout demonstrated striking loss of a specific cell type in the collecting duct rather than in other nephron segments, our paper focused on the collecting duct.

Consequently, we have removed the figure of kidney induction as well as all gene analyses that utilized Tfcp2l1induced epithelia. Fortunately, the gene analysis is unchanged. The only mention of the inductive activity of Tfcp2l1 is a brief comment in Discussion.

*5) Some of the data if very difficult to see (Figure 1P-R, Figure 2G-J, 3C, 4A, B, most of Figure 6).*

We have replaced these figures and have divided the images into their component pieces.

*In some cases, individual channels need to be shown. In others, higher magnification images would be of help.*

We now show individual confocal channels to clarify the images. We have added higher power images in Figure 2—figure supplement 1, Figure 3—figure supplement 1, Figure 4, Figure 6, Figure 7, Figure 8, Figure 9.

*6) The authors need to clarify what Notch antibody they are using in their stainings. There are discrepancies between what they say in the text, how the label their figures and what the figure legends say.*

The antibodies are listed in the Supplementary file 5. The Notch antibodies are R&D AF1057 and Abcam Ab8925. We have reviewed the figures and legends many times and hopefully there are no longer discrepancies.

*7) Figure 6A and B need to be divided up into distinct panels as each panel shows a different antibody combination from a different kidney (or at least region of the kidney). The single channel staining of pendrin is of low quality and hard to interpret. Co-stainings and/or higher mag should be shown.*

These figures have been changed and separate channels are now shown throughout the manuscript (e.g. new Figure 9). We have removed the low power image of pendrin staining substituting instead separate channels for ATPase in Figure 9.

*8) Figure 1, nuclear Tfcp staining is clear in the adults (E, F) but less so in the embryos (B, C, D). Do the authors have any better images? Also, is it clear that Tfcb is in all collecting duct cells in that levels are equal? Similarly, in 1M, the triple staining makes it hard to see whether Tfcb is in all cells. In 2E, it looks like the IC cells have more Tfcb for example.*

This is a very intriguing question concerning the regulation of Tfcp2l1 in different cell types. We changed Figure 1: E15 demonstrates an equivocal distribution of Tfcp2l1 but by E18 there is unequivocal nuclear localization co-incident with the first appearance of IC “double positive cells”. Nuclear localization of Tfcp2l1 is also clear in the adult.

We now add data from spot imaging the adult collecting duct (Figure 1). It shows that IC cells, marked by Atp6b1, have a two-fold higher expression of Tfcp2l1, than do the PC cells, marked by Aqp2. Thank you for this question; we now mention in Discussion that a difference in Tfcp2l1 levels might contribute to the differentiation of IC-PC.

*9) In Figure 3 the authors never show us the expression of Tfcp in the Adeno infected cultures, so it is really hard here to discern how much of the epithelia is due to Tfcb expression and how much is due to the supposed induction by LIF. Again, the multiple stains make it really hard to discern individual patterns.*

Tfcp2l1 is an independent epithelial inducer, which is downstream of LIF. Consequently, no LIF is needed when Ad-Tfcp2l1 is added to metanephric mesenchyme. However, we removed the data from the paper concerning the inductive activity of the transcription factor according to suggestion #4 and in order to better focus the paper on Tfcp2l1’s role in the collecting duct.

*In 3D, it is hard for any reviewer to discern whether the Tfcb ChIP-seq peaks are meaningful and whether they localize to specific regions of genes or at the TSS. From the excel sheet, individual peaks may be significant but can the authors say something about the statistical significance of location of peaks? In other words, are peaks more likely to be near TSS rather than dispersed more randomly at target genes?*

Thank you for this question. We added a better plot in the figure (new Figure 5A) and indicated the accumulation of Tfcp2l1 peaks of both up and down regulated gene near the TSS (Figure 5B). We show that ~30% of peaks are within 1kb of the TSS and 40% are accounted for within 10kb. In contrast, randomly selected genes do not demonstrate accumulation of peaks near TSS.

*10) In Figure 6, it would be nice to see Tfcb expression in the Jag1 KOs. In other words, does notch signaling alter levels of Tfcb and does this in turn reinforce more notch signaling to set boundaries between cell types?*

Thank you for this question; it is an important addition to the article. The knockout of Tfcp2l1 abolishes Jag1 staining but the deletion of Jag1 does not suppress Tfcp2l1, at least as detected by IF. We have included a new figure from Jag1 knockouts, Figure 9 and supplements, which show reversion to “double positive cells” (Atp6b1^+^and Krt8^+^) in the collecting duct, but no loss of expression of Tfcp2l1. Consequently we have proposed that Tfcp2l1 is upstream of Jag1.

On the other hand, in answer to #8, it is possible that Jag1 signaling suppresses Tfcp2l1 levels (PC cells had less Tfcp2l1; Figure 1) so there may be an additional feedback with promotes the differentiation of these cell types. We now mention this hypothesis in the Discussion.

*11) The model in Figure 7 assumes equal amounts of Tfcb, but this not clear. The transition from double positive to single positives, clearly require notch/jag1 and this depends on Tfcb, which is expressed in all cells (I think). But as notch signaling commences, do Tfcb levels remain the same or are they regulated by notch and does this then alter the amount of notch ligands to reinforce the pattern? The model assumes Hes1 and Foxi1 mediate the activation and repression, but again I am not sure Tfcb levels are equivalent in the two cell types.*

From the above #10, we think that Jag-Notch are downstream of Tfcp2l1 and hence we do not expect variation of Tfcp2l1 with Jag1 signaling. By using an in situ RNA labeling technique-pulldown assay that we developed (Unpublished), we do not find much difference in RNA content of these cells. However, to examine this question, we used spot imaging of the collecting duct and we measured a two- fold difference in immunodetectable Tfcp2l1.

We emphasized in the Discussion that there is evidence for feedback loops at the level of Foxi1 and Jag1. While our data are strongest for this hypothesis, our new data suggests that modulation of Tfcp2l1 may enhance the distinction between IC and PC as well. Thank you for pointing out this possibility.